# BCG vaccination-induced acquired control of mycobacterial growth differs from growth control preexisting to BCG vaccination

Krista E. van Meijgaarden [1,12], Wenchao Li[2,3,12], Simone J. C. F. M. Moorlag[4], Valerie A. C. M. Koeken [2,3,4,11], Hans J. P. M. Koenen[5], Leo A. B. Joosten [4,6], Annapurna Vyakarnam[7,8,9], Asma Ahmed[7,8], Srabanti Rakshit[7,8], Vasista Adiga[7,8], Tom H. M. Ottenhoff [1], Yang Li [2,3,4], Mihai G. Netea [4,10] & Simone A. Joosten [1] ✉

Bacillus Calmette-Guèrin - vaccination induces not only protection in infants and young children against severe forms of tuberculosis, but also against non-tuberculosis related all-cause mortality. To delineate different factors influencing mycobacterial growth control, here we first investigate the effects of BCG-vaccination in healthy Dutch adults. About a quarter of individuals already control BCG-growth prior to vaccination, whereas a quarter of the vaccinees acquires the capacity to control BCG upon vaccination. This leaves half of the population incapable to control BCG-growth. Single cell RNA sequencing identifies multiple processes associated with mycobacterial growth control. These data suggest (i) that already controllers employ different mechanisms to control BCG-growth than acquired controllers, and (ii) that half of the individuals fail to develop measurable growth control irrespective of BCG-vaccination. These results shed important new light on the variable immune responses to mycobacteria in humans and may impact on improved vaccination against tuberculosis and other diseases.

*Bacillus Calmette-Guèrin* (BCG) celebrated its centennial anniversary in 2021 and is still the only registered vaccine against tuberculosis (TB)[1]. BCG is an attenuated strain of *Mycobacterium bovis* that protects against severe forms of TB in infants and young children, but has limited efficacy in protecting adolescents and adults against contagious pulmonary TB, the main form of the disease[2]. BCG vaccination in infants induced variable immune responses, that did not correlate with protection in a case-control setting[3,4]. Transcriptomic analysis identified two clusters of infants with highly diverse expression profiles at 10 weeks post-BCG vaccination, likely reflecting

[1]Department of Infectious Diseases, Leiden University Medical Center, Leiden, The Netherlands. [2]Department of Computational Biology for Individualised Infection Medicine, Centre for Individualised Infection Medicine (CiiM), a joint venture between the Helmholtz-Centre for Infection Research (HZI) and the Hannover Medical School (MHH), Hannover, Germany. [3]TWINCORE, a joint venture between the Helmholtz-Centre for Infection Research (HZI) and the Hannover Medical School (MHH), Hannover, Germany. [4]Department of Internal Medicine, Radboud University Medical Center, Nijmegen, The Netherlands. [5]Department of Laboratory Medicine, Laboratory Medical Immunology, Radboud University Medical Center, Nijmegen, The Netherlands. [6]Department of Medical Genetics, Iuliu Hatieganu University of Medicine and Pharmacy, Cluj-Napoca, Romania. [7]Centre for Infectious Disease Research, Indian Institute of Science, Bangalore, India. [8]Laboratory of Human Immunology, Division of Infectious Diseases, St. John's Research Institute, Bangalore, India. [9]Department of Immunobiology, School of Immunology & Microbial Sciences, Faculty of Life Science & Medicine, King's College, London, UK. [10]Department of Immunology and Metabolism, Life and Medical Sciences Institute, University of Bonn, Bonn, Germany. [11]Present address: Research Centre Innovations in Care, Rotterdam University of Applied Sciences, Rotterdam, the Netherlands. [12]These authors contributed equally: Krista E. van Meijgaarden, Wenchao Li. ✉e-mail: s.a.joosten@lumc.nl

differentiated inflammatory responses, and suggesting divergence of responses to BCG vaccination already in neonates[4].

In addition to its partial protective efficacy against TB, BCG vaccination is well-known for its overall contribution to reduced childhood mortality beyond decreased TB mortality, likely as a result of heterologous protection against multiple pathogens during early life[5–8]. BCG vaccination increased protection against controlled challenge with a heterologous virus (yellow fever) in adults[9] and modulated responses to experimental *Plasmodium falciparum* infection, with earlier and stronger inflammatory responses but lower parasitemia[10]. BCG vaccination also augmented immune responses to subsequent vaccinations, including against SARS-CoV2[11].

One of the likely mechanisms behind heterologous protection is the induction of trained innate immunity, in vitro mimicked by β-glucan or BCG stimulation[12]. Trained monocytes are characterized by epigenetic changes, such as increased chromosomal accessibility, resulting in a response-ready state, with increased metabolic activity and secretion of hallmark cytokines IL-6, IL-1β, and TNF-α in response to pathogen sensing, e.g. via TLR triggering. The underlying epigenetic changes are dependent on changes in cellular metabolism, including increased glycolysis and cholesterol metabolism[13]. The relative increase in glycolysis, with an associated decrease in oxidative phosphorylation is called the Warburg effect[14] and was recently also described to occur in immune cells[15]. β-Glucan induced trained immunity showed a clear Warburg effect, high glycolysis, and decreased oxidative phosphorylation[16], whereas BCG-induced trained immunity resulted in both increased glycolysis and oxidative phosphorylation[13]. Moreover, host metabolic status at vaccination correlates with trained immunity induction by BCG vaccination[17].

Trained innate immunity has previously been associated with an enhanced capacity to control growth of mycobacteria[18]. Measurement of effector responses in functional assays is an important new tool to predict the actual capacity of the immune system to combat the pathogen in a relatively unbiased manner. Mycobacterial Growth Inhibition Assays (MGIA) provide the means to quantify the capacity of whole blood or peripheral blood mononuclear cells (PBMCs) to reduce the growth of *Mycobacterium tuberculosis* (Mtb) or BCG, the latter as a more accessible surrogate. A significant improvement in sensitivity has been the continuous readout in 'mycobacterial growth indicator tubes' using the BACTEC MGIT system as opposed to classical colony forming unit (CFU) counting[19,20]. Primary BCG vaccination as well as BCG-revaccination of adults were used to evaluate MGIA performance. Primary BCG vaccination in healthy adults resulted in reduced BCG growth, whereas revaccination of previously vaccinated individuals did not show enhanced mycobacterial growth control[21]. This growth control was observed at 4 and 8, but not 24 weeks post-vaccination, indicating a temporally increased capacity to control[21]. Similarly, BCG-vaccinated infants displayed increased BCG growth control at 4 months compared to non-vaccinated infants; however, at 1 year post-vaccination no differences were identified[22]. In agreement with these findings, in a previous Dutch cohort, primary BCG vaccination-induced increased growth control between weeks 4 and 12, which was lost at 1 year post-vaccination[18].

Recent exposure to individuals with contagious, active pulmonary TB, also resulted in increased mycobacterial growth control in a considerable proportion of exposed individuals. However, cells from individuals with established, presumably more remote, TB infection (TBI) did not possess the capacity to control BCG growth[18,23], supporting the notion of a temporary nature of this control, similar to the temporary BCG vaccination-induced growth control.

BCG vaccination remains an interesting model to assess and delineate host immune responses responsible for functional growth inhibition effects. The 300BCG study enrolled 325 healthy adults in the Netherlands and samples were collected before, as well as 3 months after BCG vaccination[24]. BCG vaccination reduced systemic inflammation, but at the same time enhanced cytokine responses to in vitro restimulation with non-mycobacterial stimuli such as LPS, suggesting induction of trained immunity. Males had profiles consistent with higher systemic inflammation prior to vaccination, whereas the response to vaccination was comparable to females, resulting in decreased levels of proinflammatory cytokines in the circulation[24]. In this study we continue to investigate the interactions between BCG growth control and trained immunity in response to BCG vaccination. As trained immunity is induced in approximately half of individuals vaccinated with BCG, we selected 42 individuals with strong or poor induction of trained innate immunity, defined by IL-1β production following *S. aureus* stimulation. Samples from these 42 donors before and after vaccination were tested for functional control by MGIA, all samples were subjected to scRNAseq both unstimulated and upon 4 h of LPS stimulation. This study shows that individuals with acquired growth control upon BCG vaccination, express different functional pathways in comparison with individuals that already control prior to vaccination.

## Results

### Mycobacterial growth control before and after BCG vaccination

BCG vaccination induces variable levels of protection against mycobacterial infections in different populations and individuals, but the reasons behind this remain unclear. To identify such underlying causes, we studied the capacity of PBMCs (with autologous serum) to control BCG growth as a measure of functional anti-mycobacterial responses, before and after primary BCG vaccination in a low TB endemic country (The Netherlands). Interestingly, 11 out of 42 participants (26%) already controlled BCG growth (defined as 50% reduction in CFU compared to inoculum as previously published)[18] before BCG vaccination. In the remaining participants, 8 out of 31 (26%) acquired the capacity to control BCG growth 3 months after BCG vaccination (defined as ΔlogCFU (post (V3) – pre (V1)) < −0.17, being the SD mean inoculum). By contrast, 23 out of 42 original volunteers (54% of total, 23/31 that did not control prior to vaccination (74%)) were unable to control BCG growth either before or at 3 months post-BCG vaccination (Fig. 1a).

To investigate mechanisms responsible for mycobacterial growth control, and to identify potential causative differences in the response to vaccination, individuals were grouped as 'already controlling', 'acquired control' or 'no control' (Fig. 1b). Overall, 11 (26%) of individuals controlled already prior to vaccination, 8 (19%) acquired control upon BCG vaccination and 23 (54%) did not control BCG at any time point tested. Remarkably, 9 of 11 individuals (81%) that already controlled mycobacterial growth were women, as well as 4 out of 8 (50%) in the acquired control group, while only 7 out of 23 (30%) individuals without control were women (Fig. 1c, 0.0193 $\chi^2$ test). The capacity to control BCG growth was not related to the original IL-1β response to *Staphylococcus aureus* (Supplementary Fig. 2) nor to immunological parameters of vaccine take (Supplementary Fig. 3), BCG growth control was not directly related to BCG vaccine take assessed by BCG scarring was not different between the three functional groups (Supplementary Fig. 3a) and the IFN-γ responses to heat-killed Mtb in supernatants of seven days stimulated PBMC samples as a proxy for vaccine take, was significantly increased after BCG vaccination in the group as a whole (Supplementary Fig. 3b). These IFN-γ responses were not related to the *S.aureus* nor the functional BCG growth control classifications (Supplementary Fig. 3c, d). As controls we showed that all participants responded to heat-killed bacteria, fungi or LPS with a significant increase after BCG vaccination for the *S.aureus* responders against *S.aureus*, *M.tuberculosis* and LPS stimulation (Supplementary Fig. 2a). Furthermore, IL-1β responses were not significantly different between functional groups (Supplementary Fig. 2b).

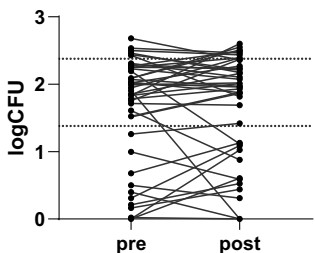

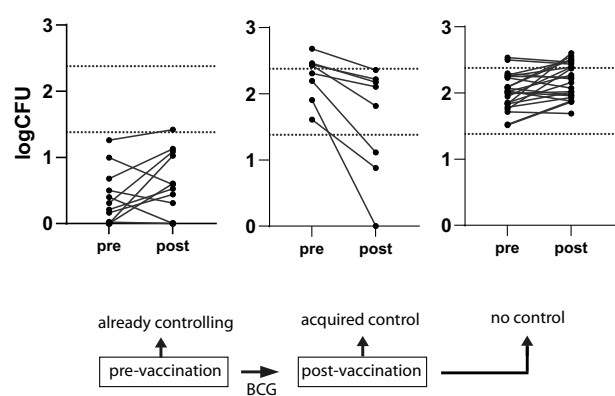

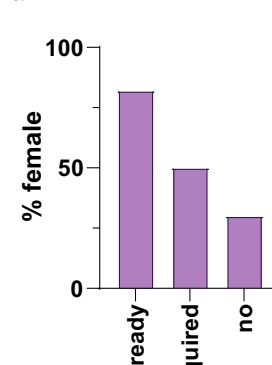

**Fig. 1 | BCG induces functional effector responses.** PBMCs were infected with live BCG, rotated 4 days, and incubated in MGIT tubes in the BACTEC machine. **a** PBMCs collected before (pre) BCG vaccination and 12 weeks after (post) vaccination were tested in the same assay. Each line represents an individual vaccinee ($n = 42$). Dotted lines represent the inoculum (logCFU 2.38) and 50% reduction of bacterial growth (logCFU 1.19). **b** The left panel displays individuals showing control of bacterial growth by logCFU reduction of more than 1 log before BCG vaccination ($n = 11$) (already). The middle panel shows individuals with a ΔlogCFU of 0.17 (SD of mean of the inoculum) or more between pre and post-vaccination sample ($n = 8$) (acquired). In the right panel individuals that do not show growth control at either of the time points tested are depicted ($n = 23$) (no control). **c** % female (Y-axis) over the functional groups on the X-axis.

## Cyto-/chemokine profiles vary with control of mycobacterial growth

The 3 functional groups (already controllers, acquired controllers, no controllers) were used to identify differences in immune cell subset composition, circulating cytokine/chemokine profiles, as well as

cytokine/chemokine production in response to BCG stimulation. The functional groups did not differ in frequencies of major immune cell subsets, either before or after vaccination (Fig. 2a, gating strategies in Supplementary Fig. 4b). Although no major subsets were associated with the three functional profiles, we did observe branching in the heatmap, separating no control from the two other groups, and, interestingly, linking acquired controllers prior to vaccination more closely to the no controllers. Samples from acquired controllers (at the post-vaccination time point) were most unique in composition, in particular in multiple B-cell subsets.

Analyses of soluble analytes in plasma samples resulted in branching of already controllers into a separate cluster both before and after vaccination. Acquired controllers branched separately from no controllers at pre and post-vaccination time points as well (Fig. 2b). Pre-vaccination samples of acquired controllers grouped with samples that lacked control more closely than with already controlling samples (Fig. 2a). Although clustering was in line with the functional groupings, it was difficult to associate individual markers with the control status in these plasma samples. The plasma soluble analyte profile from individuals that did not control BCG growth was very similar prior to and postBCG vaccination.

Previously, secreted chemokines in the supernatants after 96 h of BCG stimulation were linked to functional profiles[18], therefore we determined also the cytokine and chemokine concentrations upon BCG stimulation. Similar to ex-vivo plasma concentrations, production of cytokines and chemokines upon 96 h of in vitro BCG stimulation separated already controllers from individuals that acquired control and those that did not control at either time point (Fig. 2c). Relative production of CCL17, CCL1, IL-4, CCL27 and CX3CL1 was high in individuals that already controlled BCG growth before vaccination (Supplementary Fig. 5a). Interestingly, these concentrations were not increased upon vaccination in individuals that acquired control, suggesting that distinct, non-overlapping immunological pathways are activated by in vitro BCG stimulation. Individuals with mycobacterial control, both in already controlling and in acquired control groups, had relatively low concentrations of CCL21 and CCL25 (both chemokines are associated with inflammatory disorders) upon BCG stimulation (Fig. 2c, Supplementary Fig. 5b). In unstimulated plasma, CCL25 was also lower in already controllers (Fig. 2b). CCL25 was also previously identified to be higher in men compared to women prior to vaccination[24], which is in line with our finding of an increased representation of women in the already controllers.

## BCG growth control associated with increased monocyte populations

In a completely independent cohort of individuals from Bangalore, India, that received a BCG-revaccination we identified very similar patterns, confirming the data in the 300BCG cohort of European ancestry. The primary BCG vaccination was administered directly at birth and considering the age of the participants (average age 20.2 years with a range of 18–24 years), this was long before inclusion into the revaccination study. All individuals were IGRA negative at the time point of inclusion. Participants received the BCG Russia vaccine as BCG-revaccination and the IFN-γ response to BCG in a 12 h stimulation of whole blood was used as a measure of vaccine take. We observed a significant increase in the number of IFN-γ⁺ CD4⁺ T-cells in the samples taken 10–12 weeks post-vaccination compared to pre-revaccination (Supplementary Fig. 3e, f).

In this BCG-revaccination study, we identified 10/57 (18%) individuals that already controlled BCG growth prior to revaccination or at inclusion in the control arm, 11/23 (43%) individuals that acquired control upon BCG-revaccination and 12/23 (52%) individuals that did not control BCG at either time point despite BCG-revaccination (Fig. 2d). This pattern is very comparable to that observed in the Dutch cohort (Fig. 1b). Sex distribution in the Indian cohort was balanced

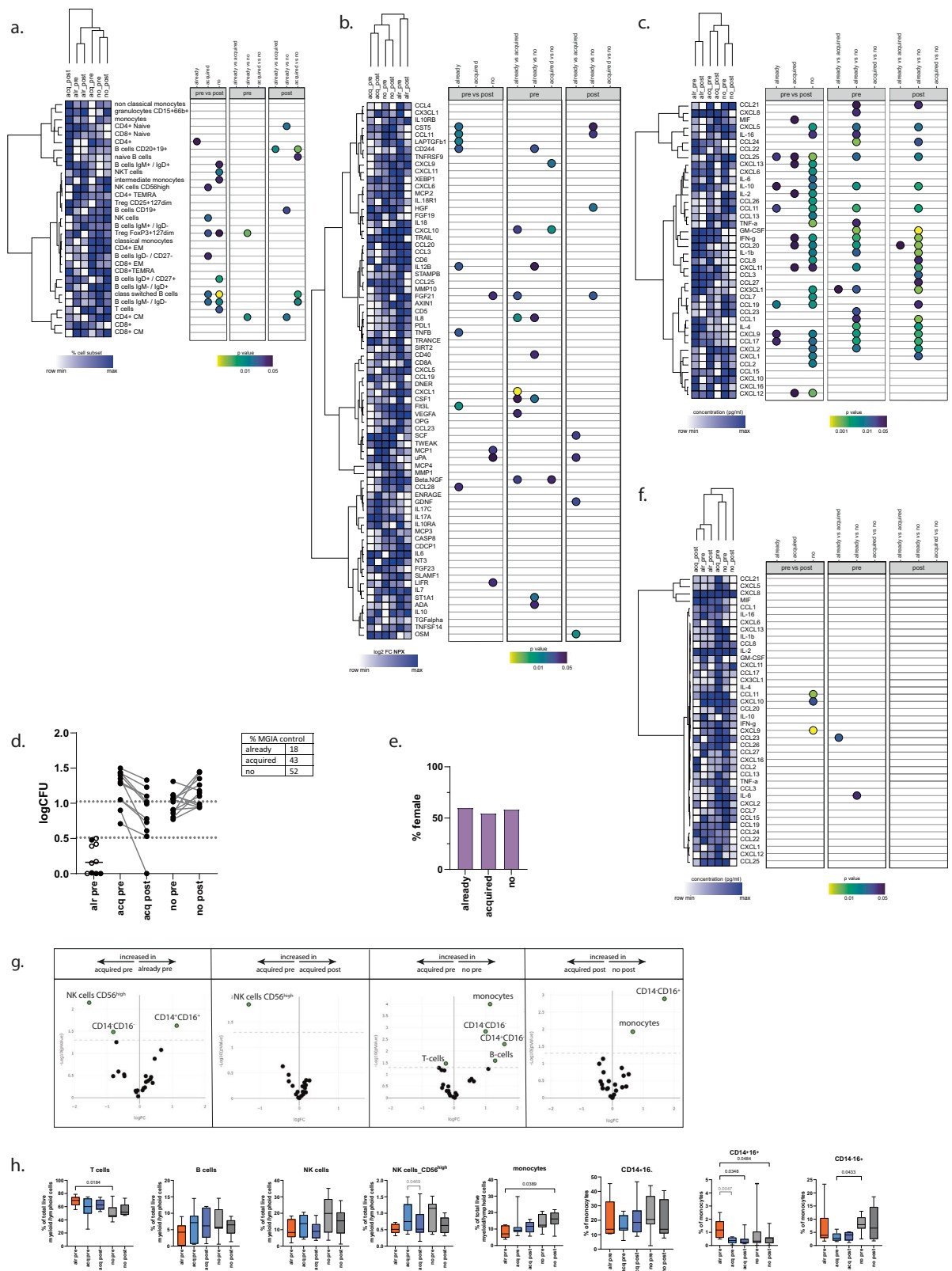

across the three functional groups (Fig. 2e). Cytokines and chemokines were measured upon 96 h of BCG stimulation. Samples from volunteers that already controlled mycobacterial growth prior to revaccination and those that acquired control upon BCG-revaccination again clustered closest together (Fig. 2f).

In this cohort, more extensive flow cytometric phenotyping was performed using a spectral flow panel[25]. Comparative analyses

identified limited differences in NK and monocyte subsets between the functional groups (Fig. 2g, h). The already controllers displayed increased frequencies of CD14+CD16+ intermediate monocytes in comparison with individuals that acquired control by revaccination. Classical CD14+CD16− monocytes showed a large variance in all groups and there was no correlation between control state and frequency of this monocyte subset. There was a trend towards increased

**Fig. 2 | Different cytokine/chemokine profiles for already controllers of BCG growth.** Heatmaps were created using hierarchical clustering on rows and columns with Euclidean distance and average linkage method using Morpheus software (Broad Institute). Color scaling is relative within each row, dark blue as maximum, white for minimum values. Individuals were grouped based on their capacity to control BCG growth, medians were used. Statistical analysis (two-sided) was performed for within-group comparisons by a Mann–Whitney matched-pairs-test followed by Benjamini–Hochberg multiple test correction, between-group comparisons by Mann–Whitney U-tests and Benjamini–Hochberg. Significant $p$-values were depicted as dots with the level of significance according to color scaling. **a** PBMCs were analyzed using flow cytometry, subset frequencies are displayed pre and post-BCG vaccination. **b** Circulating inflammatory markers were determined in plasma collected before and 12 weeks post-BCG vaccination. **c** Chemo- and cytokine levels were quantified in supernatants from 96 h BCG stimulation samples. **d** PBMCs from an independent validation cohort of Indian adults with ($n = 27$) or without ($n = 30$) BCG-revaccination were tested in the MGIA and plotted according to their capacity to control BCG growth. Dotted lines represent the inoculum at 1.025 logCFU with 50% reduction at 0.513 logCFU. Individuals with (black symbols) or without BCG-revaccination (open symbols). **e** Distribution of females over functional groups. **f** Chemo- and cytokines were quantified in supernatants from 96 h BCG stimulation samples. **g** Immunophenotyping was performed if sufficient PBMCs were available after performing the functional assay (already, $n = 8$; acquired, $n = 8$; and no control, $n = 8$). Cell subsets that are significantly different are shown in green in the EdgeR volcano plots. Datasets were exported for univariate analysis between all functional groups (**h**) and tested by Kruskal–Wallis with Dunn's multiple test correction (already, $n = 8$; acquired, $n = 8$; and no control, $n = 8$). Within-group testing was performed by a Wilcoxon matched-pairs test (acquired; $n = 7$), the comparison between already and acquired control pre-vaccination for the CD14$^+$CD16$^+$ subset was tested by Mann–Whitney ($n = 8$). Box and whiskers summarize the median (thick line), 5–95 percentile (box), and minimum-maximum (whiskers).

non-classical monocytes (CD14$^-$CD16$^+$) in already controllers, compatible with their previously described role in BCG growth control[18]. Already controllers also had an increased number of T-cells compared to those that did not control (Fig. 2g, h). CD56$^{hi}$ NK cell frequencies were decreased upon acquisition of control, possibly reflecting reduced activation of NK cells in samples with control (Fig. 2g, h). Together, these data from the Dutch primary vaccination cohort as well as from the Indian BCG-revaccination cohort suggest that already controllers have different characteristics as compared to acquired controllers upon BCG (re)vaccination.

## Genes associated with metabolic activity in already control
Single-cell RNA sequencing of Dutch 300BCG PBMC samples stimulated for 4 h with LPS (as heterologous innate immune activator), or an unstimulated control sample, was performed to associate transcriptome profiles with functional BCG growth control phenotypes. Individuals were categorized based on the functional MGIA assay and differentially expressed genes determined before and after BCG vaccination.

The comparison between individuals that already control BCG growth before vaccination to those that were not able to control BCG at any of the tested time points revealed similar numbers, respectively 1516 and 1577, DEGs (differentially expressed genes) in both unstimulated as well as LPS-stimulated samples (Fig. 3a). All DEG lists are provided in the source data file. DEGs were identified in all deconvoluted cell subsets at approximately similar rates (B-cells, CD4$^+$ T-cells, CD8$^+$ T-cells, NK cells, and monocytes), indicating a generic difference in immune status rather than activation of a specific cellular subset (Fig. 3a). Monocyte gene-expression patterns in particular were very different between the groups, with both genes up and down-regulated in conditions with mycobacterial growth control indicating active up and downregulation (Fig. 3b). Subsequent integration of these DEGs into biological processes revealed that cells isolated from already controllers before vaccination displayed strongly upregulated metabolic processes, including oxidative phosphorylation, ATP synthesis, and electron transport, as well as mitochondrial respiration (Fig. 3c). Interestingly, these pathways, related to increased metabolic activity, were identified in all immune cell subsets studied, suggesting increased metabolic activity in multiple cell subsets. Additional pathways identified were related to type I IFN responses and defense against viruses, suggesting a true response-ready status of the immune players in samples of already controllers (Fig. 3c). Biological processes identified by DEGs detected upon TLR4 stimulation by LPS revealed very similar metabolic activation, including ATP synthesis and oxidative phosphorylation related processes in most subsets, but to a lesser extent in monocytes. This is likely due to the fact that monocytes become most strongly activated during the 4 h of LPS stimulation and the time point selected

is less optimal. Additional pathways that were upregulated in already controllers upon LPS stimulation were responses to proinflammatory cytokines including IFN-γ, IL-1, and TNF-α, and regulators of lymphocyte proliferation, reflecting a strong proinflammatory response (Fig. 3d).

We further explored the RNAseq data in deconvoluted monocyte subsets based on transcription profiles and used these subsets to again probe differences in biological processes (Fig. 3e). Monocyte deconvolution was only successful in unstimulated samples. Both classical and non-classical monocytes from already controllers at inclusion showed increased expression of genes associated with cellular metabolism compared to monocytes from individuals that lacked control (Fig. 3e), suggesting again that the capacity to already control is associated with increased metabolic activity in monocytes.

Comparison of gene-expression patterns prior to vaccination between already controllers before vaccination to acquired controllers upon BCG vaccination revealed similar DEG numbers, 1487 and 2124 genes, respectively, for both unstimulated and LPS-stimulated samples. Distribution over deconvoluted cell subsets was similar as in the comparison with individuals that lacked control (Supplementary Fig. 6a). Similarly, monocytes in already controllers had both significant numbers of up- and down-regulated genes compared to samples from no controllers (Supplementary Fig. 6b). Biological processes between the already controllers and acquired controllers in comparison with the group unable to control mycobacterial growth were very similar (Supplementary Fig. 6c, unstimulated and LPS stimulated). Already controllers had gene-expression patterns consistent with antimicrobial effector responses, antigen processing and ATP synthesis, and electron transport. These patterns strongly agree with the findings in the comparison of already controlling compared to non-controlling individuals.

## Acquired control induces metabolic pathways and effector responses
Comparing pre- vs post-vaccination gene-expression profiles revealed a strong increase in DEG numbers in the acquired controllers (Fig. 4a, 191 DEGs for unstimulated and 685 DEGs for LPS-stimulated samples), but no such changes in individuals that did not control BCG growth at either time point (Fig. 4b, 32 and 42 DEGs for unstimulated and LPS-stimulated samples respectively). Already controllers showed minimal changes at the post-vaccination time point (Supplementary Fig. 7a). Notably, in acquired controllers most DEGs were detected in monocytes, 529 DEGs, in particular upon LPS activation, with some changes in CD4$^+$ and CD8$^+$ T-cells, 57 and 46 DEGs, but only a small number of genes changed in the other cell subsets upon BCG vaccination, 20 DEGs in B-cells and 33 DEGs in NK cells. Comparison of acquired controllers with no controllers post-vaccination revealed a high number of DEGs, in particular upon LPS stimulation (Fig. 4c, 460 DEGs for

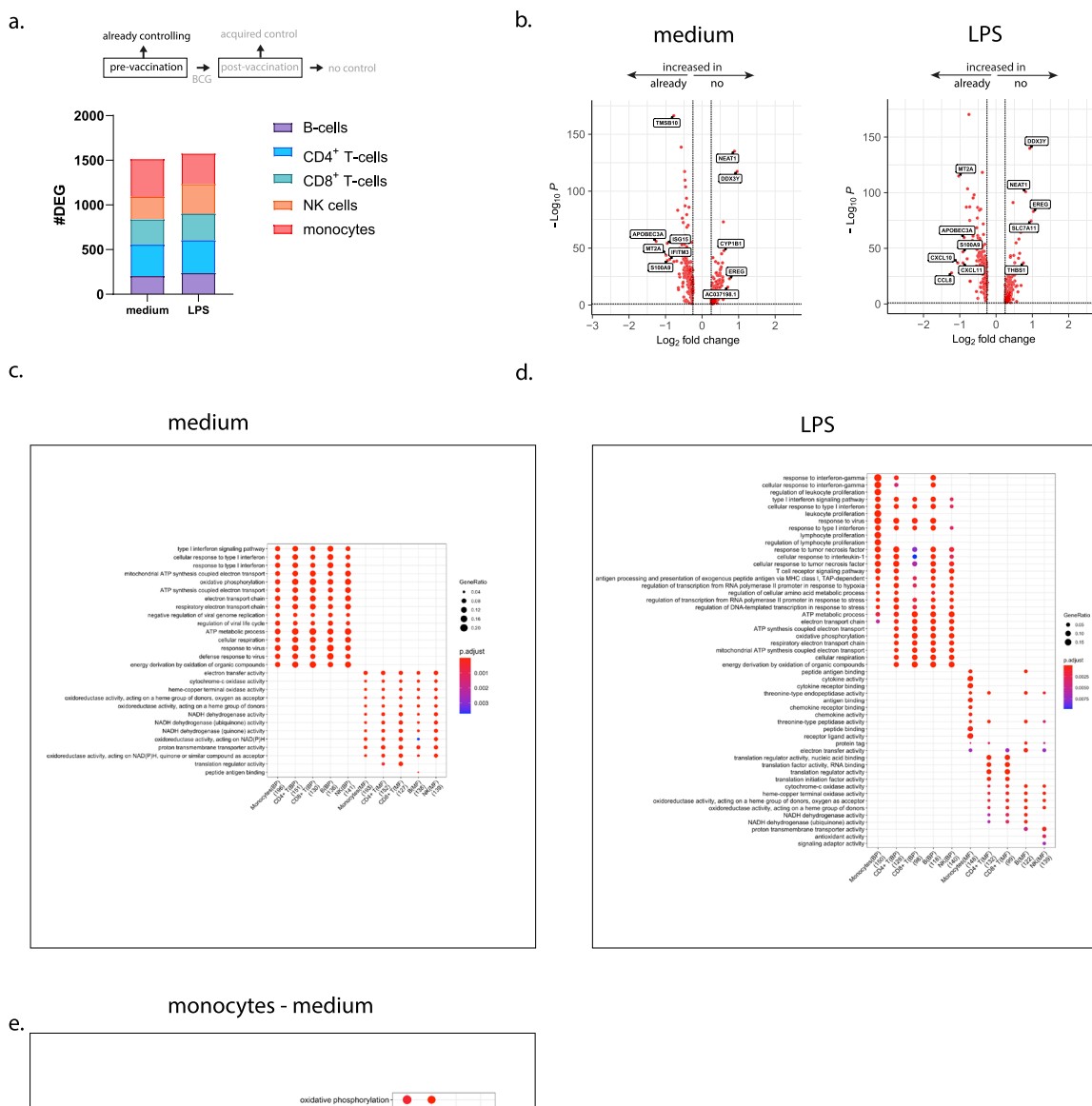

unstimulated and 1294 for LPS-stimulated samples). Compared to individuals incapable of controlling mycobacterial growth, acquired control displayed the largest DEG numbers in CD8[+] T-cells and monocytes, in particular upon LPS stimulation (Fig. 4c, 434 and 635 DEGs respectively). Samples of acquired controllers showed both up and downregulation of genes within the monocytes, in particular upon LPS stimulation (Fig. 4d). Whereas changes in monocytes were

abundant, more subtle changes were observed in other subsets like CD4[+] T-cells. Individuals that did not acquire control post-BCG vaccination showed almost no changes in transcriptional profiles (Fig. 4d). Post-vaccination, samples that acquired the capacity to control BCG growth showed up as well as downregulation of many genes in monocytes and CD8[+] T-cells compared to individuals that did not acquire the capacity to control (Fig. 4e). BCG vaccination in acquired

**Fig. 3 | Differential gene expression in already controllers prior to BCG vaccination.** PBMCs were stimulated with medium or LPS for 4 h and processed for single-cell RNAseq analysis. **a** Stacked bars represent the number of differentially expressed genes after deconvolution of cellular subsets, between already controllers vs. no controllers prior to BCG vaccination for unstimulated and LPS-stimulated samples, colors indicate the respective subsets. **b** Scatter-volcano plots showing significance (-$\log_{10}P$) versus magnitude of change (Log$_2$ fold change) of the differentially expressed genes for both the unstimulated medium (left plot) vs. the LPS-stimulated monocyte subset (right plot). Statistical testing was performed by Wilcoxon Rank Sum test (two-sided) and genes were considered significant when expressed in at least 10% of the cells and an adjusted *p*-value < 0.05 after Benjamini–Hochberg correction. Genes with a log fold change <0.25 are increased

in already controllers, genes with a log fold change >0.25 increased in no controllers after BCG vaccination. **c** Pathway analysis showing the biological processes (BP) and molecular functions (MF) related to the differentially expressed genes, split over the different cell subsets for the unstimulated samples, for LPS-stimulated samples (**d**), and the unstimulated monocyte subsets solely dividing them in non-classical monocytes (ncM) and classical monocytes (cM) (**e**). Dot colors represents the adjusted p-values and dot sizes the gene ratios. For every subset the number of genes involved is shown between brackets and for the enrichment test, significant gene sets were subjected to enrichGO and enrichKEGG function (pAdjustMethod = "BH", qvalueCutoff = 0.05) and pathways with BH correction (FDR) < 0.05 were considered significant.

controllers induced gene-expression patterns compatible with increased responses to IFN-γ, type I interferons, increased antigen presentation and strongly enhanced migratory markers in mostly the lymphocyte subsets when comparing pre vs post-vaccination time points (Fig. 4f). A direct comparison of the acquired controllers with the no controllers at the post-vaccination time point showed that acquired control was associated with extensive metabolic reprogramming of monocytes, and increased expression of genes associated with antigen processing and presentation (Fig. 4g). In addition, pathways associated with proliferation and migration of multiple subsets showed increased expression in acquired control also when compared with no control (Fig. 4g). Together these data show a clear response to BCG vaccination in the group that acquired control, but a complete lack of response in individuals that were not capable of controlling BCG growth in the functional assay.

Metabolic reprogramming, consistent with trained innate immunity was observed when comparing acquired controllers with no controllers at both time points, but not when comparing the pre and post-vaccination time points of the acquired controllers (Fig. 4f). Since the group size of the acquired control group was rather limited (*n* = 8), we decided to re-analyze the data from this group without the stringent FDR correction applied to all other analyses. In the absence of FDR correction, acquired controllers also showed induction of metabolic rewiring consistent with the induction of trained immunity in the pre- postvaccination comparison, confirming BCG vaccination-induced metabolic rewiring in individuals that also acquired functional control.

## Control before vaccination differs from vaccination-induced control

In the previous analyses the capacity to control mycobacterial growth at the time of inclusion ("already controllers") was associated with gene-expression profiles that differed in all immune subsets compared to samples that lacked control, while samples that were able to control mycobacteria following BCG vaccination ("acquired controllers") showed predominantly differential monocyte gene expression. Therefore, we performed a direct comparison of the samples that controlled BCG growth, i.e. pre-vaccination samples from already controllers with post-vaccination samples from acquired controllers. The effect of BCG vaccination on gene-expression patterns was minimal in the already controllers, such that we decided to use their pre-vaccination time points in the comparisons described below in an attempt to define this mechanism of control in the already controllers (Supplementary Fig. 7b).

When comparing those samples that are functionally similar, i.e. samples from already controllers prior to vaccination with samples from acquired controllers upon BCG vaccination, a large number of DEGs was identified both in unstimulated (907) and LPS-stimulated (1278) samples, and in many immune cell subsets, suggesting involvement of various immune cell populations (Fig. 5a). Monocytes of these groups showed considerably different patterns, with 255 (unstimulated) and 322 (LPS) genes specifically upregulated in

acquired vs. already controllers' monocytes (Fig. 5b). Further assessment of the pathways associated with the differentially expressed genes showed that samples with control prior to vaccination abundantly expressed type I IFN related pathways, pathways related to increased metabolic activity, and anti-pathogenic (anti-viral) responses. Molecular functions associated with already control of mycobacterial growth included metabolic paths, as well as cytokine and chemokine signaling (Fig. 5c). In contrast, acquired controllers expressed genes associated with cell differentiation, cell signaling, and cellular-mediated killing (Fig. 5d). Inflammatory processes in the acquired controllers were related to IL-1 and chemokines, rather than type I or II IFN; chemokine receptor binding and activity related pathways were also strongly upregulated (Fig. 5d). Although changes upon BCG vaccination were minimal in already controllers (Supplementary Fig. 7), nevertheless we did compare the differences in samples at the post-vaccination time point in already controllers with acquired controllers by vaccination. As expected numbers of DEGs were comparable to those in Fig. 5a and also the biological processes associated with these differentially expressed genes were very comparable (Supplementary Fig. 8), confirming that different pathways are activated in samples of already controllers compared with acquire controllers upon vaccination.

Venn diagrams were made to compare the genes that were differently expressed within the acquired controllers pre and post-vaccination with the DEGs from the comparison of already controllers (pre-vaccination) vs. the acquired controllers (post-vaccination) (Fig. 5e). BCG vaccination in the acquired controllers induced expression of many genes, but almost exclusively in the monocytes, especially after stimulation with LPS. However, only a limited proportion of these genes (158/529 genes = ~30%) overlapped with the DEGs identified in samples with already mycobacterial growth control (Fig. 5e). This again confirms that BCG vaccination in individuals that acquired control activated other gene-expression profiles compared to already controllers of BCG at inclusion.

Although the different pathways in Fig. 5c, d already indicate a divergence in mechanisms employed to control BCG growth between already and acquired controllers, we also performed an Ingenuity Pathway comparative analysis to rank the differences. In this analysis, genes upregulated in already controllers were directly compared to genes upregulated in samples from acquired controllers. In already controllers, among the top pathways that differed from acquired controllers were interferon signaling, oxidative phosphorylation, mitochondrial dysfunction, and pathways related to macrophage activation and cytokine storms (Fig. 5f). Ranking on acquired control resulted in top processes including glucocorticoid signaling, crosstalk between cell subsets and Th1 and Th2 pathway activation (Supplementary Fig. 9a).

Analysis of the metabolic pathways revealed that already controllers expressed genes associated with oxidative phosphorylation and mitochondrial dysfunction, whereas samples from acquired controllers expressed increased levels of genes associated with glycolysis as well as genes involved in pyruvate fermentation to lactate, a

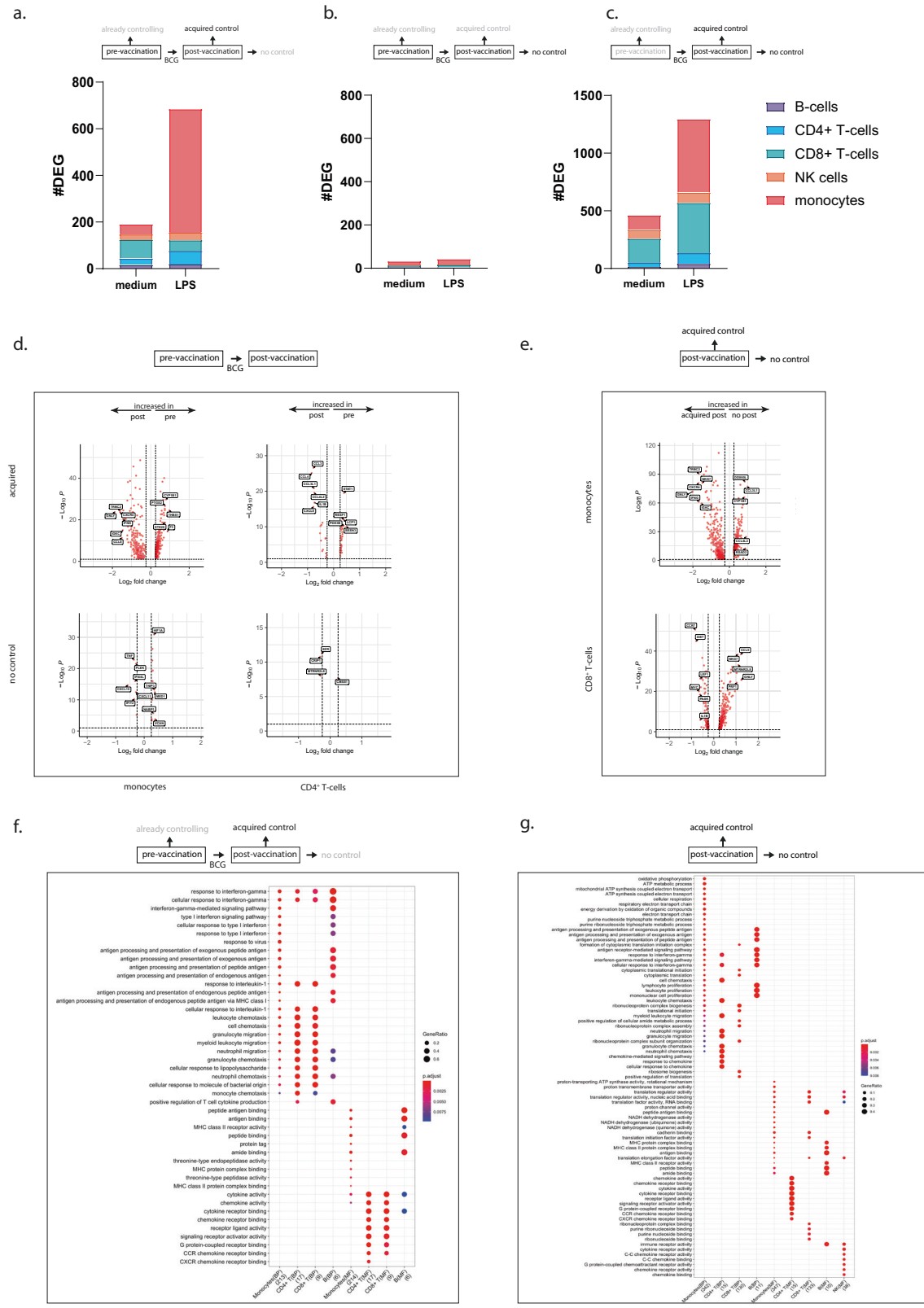

phenomenon known as the Warburg effect. In line with these findings, sirtuin pathways were upregulated in already controlling samples, whereas HIF-1α signaling was increased in samples with acquired control. Thus, metabolic rewiring is opposite in samples from already vs. acquired controllers, with oxidative phosphorylation most dominant in already controllers and glycolysis as most dominant in acquired controllers (Supplementary Fig. 9b).

This, oxidative phosphorylation was identified as one of the dominant pathways increased in already controlling samples, and indeed many components of the electron transport chain showed increased expression in samples of already controllers prior to vaccination, including those of the rate-limiting complex V (Fig. 5g). Very similarly, many components of type I interferon signaling were abundantly expressed in samples that control prior to vaccination

**Fig. 4 | Mechanism of control induced by BCG vaccination.** PBMCs were stimulated with medium or LPS for 4 h and processed for single-cell RNAseq analysis. Stacked bars represent the differentially expressed number of genes after deconvolution of cell subsets for individuals who acquired BCG control (**a**), individuals who did not acquire BCG control upon BCG vaccination (**b**) and the comparison between acquired controllers to no controllers (**c**) for unstimulated (medium) and LPS-stimulated samples, colors indicate the respective subsets. **d** Scatter-volcano plots showing significance (-log$_{10}$P) versus magnitude of change (Log$_2$ fold change) of the differentially expressed genes for the LPS-stimulated samples pre (log fold change >0.25) and post (log fold change <0.25) BCG vaccination for both the acquired controllers and the no controllers for monocytes and CD4$^+$ T-cells, **e** the LPS-stimulated samples after BCG vaccination in the acquired controllers (log fold change <0.25) and the no controllers (log fold change >0.25) for monocytes and CD8$^+$ T-cells. Statistical testing was performed by Wilcoxon Rank Sum test (two-sided) and genes were considered significant when expressed in at least 10% of the cells and an adjusted p-value < 0.05 after Benjamini–Hochberg correction. **f** Pathway analysis of biological processes (BP) and molecular functions (MF) for the upregulated genes in the LPS-stimulated samples within the acquired controllers after BCG vaccination, related to the differentially expressed genes in (**a**). **g** Pathways associated with the upregulated genes in the LPS-stimulated samples after BCG vaccination when comparing the acquired controllers to the no controllers. Dot color represents adjusted p-values and dot sizes the gene ratios. For every subset the number of genes involved is shown between brackets and for the enrichment test, significant gene sets were subjected to enrichGO and enrichKEGG function (pAdjustMethod = "BH", qvalueCutoff = 0.05) and pathways with BH correction (FDR) < 0.05 were considered significant.

compared to those that acquired the capacity to control upon vaccination (Fig. 5g).

## Discussion

Immunity induced by BCG vaccination provides partial protection against TB, and also against heterologous infections, particularly in young infants. Little is known regarding the mechanism(s) of protection employed by BCG, although innate immune reprogramming (also termed *trained innate immunity*) has been associated with the increased response-ready state against heterologous stimuli. Here we first determined the functional capacity of PBMCs to control growth of BCG before and after BCG vaccination, and subsequently categorized individuals into three groups: individuals already controlling mycobacterial growth before vaccination; individuals who acquire the ability to control mycobacterial growth control upon BCG vaccination; and individuals that were unable to control mycobacterial growth pre as well as post BCG vaccination. At inclusion, already controllers had different gene-expression profiles compared to no controllers or acquired controllers upon vaccination, and this differential expression was observed in all immune cell subsets investigated. Upon vaccination, abundant changes in gene-expression profiles were observed in individuals that acquired the capacity to control mycobacteria after BCG vaccination, but these changes were almost exclusively in monocytes. Although we cannot completely exclude classical macrophage activation, the fact that we performed our analysis at 3 months post-vaccination supports the idea that the monocytic cells are reprogrammed rather than only activated. Comparative analysis yielded multiple pathways that were differentially expressed between both groups with already vs. acquired control, both at the immunological and at the metabolic level, indicating that different pathways of host responsiveness can be involved in mediating mycobacterial growth control.

Cellular metabolism is associated with the functional state of immune cells[26]; activated T-cells have high rates of both glycolysis and oxidative phosphorylation, whereas memory T-cells are more dependent on lipid synthesis[27,28]. In macrophages, a proinflammatory phenotype is associated with glycolysis and impaired oxidative phosphorylation, whereas more anti-inflammatory macrophages rely on oxidative phosphorylation and beta-oxidation[26,29,30]. Oxidative phosphorylation and glycolysis are tightly balanced, with HIF-1α a as key regulator promoting glycolysis and inhibiting oxidative phosphorylation. HIF-1α in turn is balanced by sirtuins, which inhibit HIF-1α and thereby glycolysis, thus increased expression of sirtuins is frequently linked to efficient energy generation via oxidative phosphorylation. LPS stimulation can activate sirtuin 1 in macrophages, thereby skewing towards more anti-inflammatory states and accompanying fatty acid oxidation[31]. Similarly, in trained monocytes sirtuin 1 expression was decreased[16]; since sirtuins function as histone deacetylases, this may directly result in the increased histone acetylation observed in trained monocytes[16]. In β-glucan as well as BCG-trained monocytes glycolysis was highly activated and accompanied by

upregulation of genes in the HIF-1α and mTOR signaling pathways[13,16]. HIF-1α signaling is crucial for the induction of trained immunity[16,32]. Here, we observed increased expression of genes related to glycolysis and cellular respiration consistent with trained innate immunity in acquired control monocytes. Pathway analysis not only identified genes directly linked to glycolysis and its rate-limiting regulator, the HIF-1α pathway, but also genes involved in lactate production, suggestive of energy metabolism moving away from efficient oxidative phosphorylation to glycolysis and fermentation (NADH regeneration by electron transport to pyruvate, shifting from oxidative phosphorylation to fermentation), also known as the Warburg effect[14,16,33]. In addition, cells from individuals that acquired mycobacterial growth control upon BCG vaccination had activated mostly antimicrobial gene paths, as well as many pathways of cellular communication, with responses to Th1 and Th2 cytokines and to molecules of microbial origin. In contrast, already controllers prior to vaccination showed increased oxidative phosphorylation, mitochondrial dysfunction, and responses to type I interferons, as well some microbially triggered pathways. We also identified increased expression of genes in the sirtuin pathway directing energy metabolism to most efficient oxidative phosphorylation pathways.

BCG-induced trained immunity thus seems to be linked to increased glycolytic activity, which is also evident from several other studies on BCG vaccination cohorts. Firstly, early transcriptomic signatures obtained in blood samples collected at 10 weeks of age from neonates vaccinated with BCG at birth identified 2 clusters of responses, which were not linked to protection to TB or lack thereof[4]. One of these clusters (cluster 2), both in stimulated and unstimulated conditions, was enriched for genes linked to oxidative phosphorylation and glucose metabolism, suggesting that also in neonatal BCG vaccination metabolic reprogramming occurs in some but not all individuals[4]. BCG vaccination in healthy adult volunteers also suggested increased glycolytic activity, as stimulation of PBMCs resulted in increased lactate secretion, a hallmark of increased glycolysis[13]. Genes related to glycolysis and glucose metabolism were increased in the circulation of individuals previously vaccinated with BCG and now rechallenged with BCG[34]. In addition, IFN-γ and IL-17 were increased in those individuals and correlated directly with reduced bacterial loads at the challenge site[34]. The link with IFN-γ is interesting, since it has been shown that GAPDH, a glycolysis enzyme, can bind the promotor region of *IFNG* and decrease IFN-γ secretion. However if GAPDH is used for glycolysis, T-cells secrete more IFN-γ, an important effector molecule in mycobacterial infections[35]. Employing similar mechanisms, GAPDH may also regulate TNF production in myeloid cells[36].

Already control of mycobacterial growth was observed in approximately a quarter of Dutch donors, which is an unexpectedly high proportion in this very low burden country. Already control was also observed in 17% of Indian donors, who were nevertheless vaccinated a birth. We have previously described that recent exposure to patients with contagious active pulmonary TB resulted in a high frequency of individuals with growth control capacity[18]. However, with an

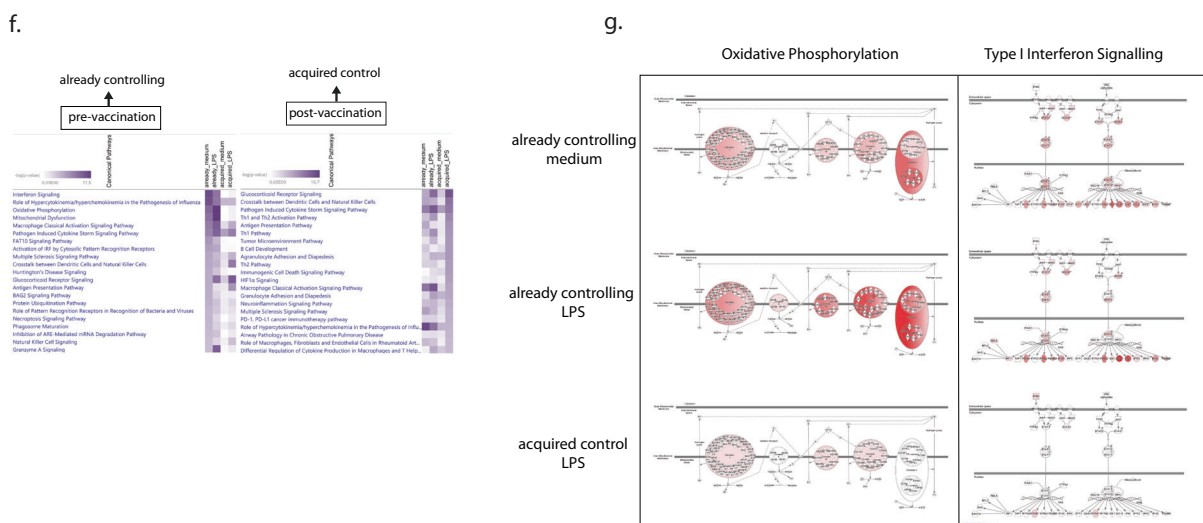

annual TB incidence of ~4.7 per 100,000, mostly amongst foreign-born immigrants, it is highly unlikely that 25% of the Dutch population that was recruited into the study was recently exposed to a case of active pulmonary TB. Upon BCG vaccination, control of mycobacterial growth lasted less than 1 year, though it is unknown whether control upon recent TB exposure (or exposure to other unknown microbes) is also of limited duration. It would be of interest to perform long-term

follow-up of individuals that control already in the absence of vaccination, to study the longevity of this phenotype as well as possible underlying mechanisms, including host genetic associations. Low dose Mtb challenge in Rhesus and Cynomolgus macaques also resulted in induction of increased growth control by PBMCs approximately 6 weeks after infection, which persisted at 12 weeks post infection[37], supporting our previous work that not only vaccination but also

**Fig. 5 | Already controllers show different gene-expression signatures compared to acquired controllers upon BCG vaccination.** PBMCs were stimulated with medium or LPS for 4 h and processed for scRNAseq analysis. **a** Stacked bars represent numbers of differentially expressed genes after deconvolution of cell subsets, between already controllers vs. acquired controllers upon BCG vaccination for unstimulated and LPS-stimulated samples, colors indicate respective subsets. **b** Scatter-volcano plot showing significance ($-\log_{10}P$) versus magnitude of change ($\log_2$ fold change) of differentially expressed genes for LPS-stimulated samples before and after BCG vaccination for already controllers and acquired controllers. Genes with a log fold change <0.25 are increased in samples post-BCG vaccination in acquired controllers, genes with a log fold change >0.25 are increased in already controllers before vaccination. Statistical testing by Wilcoxon Rank Sum test (two-sided) and significant when expressed in >10% of cells with adjusted $p$-value < 0.05 after Benjamini−Hochberg correction. **c** Pathway analysis showing biological processes (BP) and molecular functions (MF) for different cell subsets for LPS-stimulated samples representing pathways upregulated in samples of already controllers prior to BCG vaccination and **d** pathways upregulated in samples that

acquired control upon BCG vaccination. Dot color representing adjusted $p$-values, dot sizes gene ratios, number of genes involved between brackets. For the enrichment test, significant gene sets were subjected to enrichGO and enrichKEGG function (pAdjustMethod = "BH", qvalueCutoff = 0.05), and pathways with BH correction (FDR) < 0.05 were significant. **e** Venn diagrams comparing DEGs of acquired controllers upon BCG vaccination (pre and post, blue) to those of already controllers vs. acquired controllers post-BCG vaccination (green), for unstimulated (light shades) and LPS-stimulated samples (darker shades). **f** Heatmap showing the most enriched canonical pathways in a comparison analysis based on differentially expressed genes involved in BCG growth control, with purple shading representing $-\log(p\text{-value})$ ranked on already controllers prior to vaccination (left) or acquired controllers after BCG vaccination (right). **g** Canonical pathway diagram of Oxidative Phosphorylation and type I Interferon Signaling networks for different BCG controlling groups with or without LPS stimulation. Upregulated genes (red), unchanged genes (gray), unidentified genes (white), darker color equals higher relative expression.

infection may enhance functional effector responses. Alternative to exposure to Mtb, exposure to other, less pathogenic, non-tuberculous mycobacteria (NTM) such as *M.avium, M.kansassi* or *M.marinum* might induce similar responses, although this is as yet speculative. Of note, the proportion of controllers prior to vaccination in this study is similar to what we previously observed in a cohort of healthy Dutch non-BCG-vaccinated blood bank donors, where ~30% of healthy donors were able to control BCG growth[18]. The occurrence of similar frequencies (25-30%) of non-BCG-vaccinated healthy individuals that control BCG in two independent cohorts in a low TB endemic country suggests that there are also other factors that may activate the state of control, or that some individuals have the already controlling capacity as a result of e.g. genetic variation or (recent) exposure to other pathogens, including NTM, but these data are unavailable for the current cohorts. One peculiar finding in the already controllers prior to vaccination was the high frequency of women. Previously, in the same cohort, concentrations of circulating proinflammatory cytokines were reported to be increased in men, and these concentrations were subsequently dampened by BCG vaccination[24]. Perhaps this is a reflection of the more resting state of cells in samples that already control, which occurs most frequently in women. In addition, host metabolism at the time of vaccination correlated with trained immunity induced by vaccination[17]. Thus, the host inflammatory and metabolic state may also influence vaccine-induced effector responses.

Finally, about half of the Dutch donors vaccinated with BCG did not acquire the capacity to control mycobacterial growth, and showed hardly any changes in transcriptome profiles. This could suggest poor responsiveness to BCG vaccination, possibly through very rapid innate clearance of BCG without induction of innate or adaptive memory responses. However, the available proxies of vaccine take, for the Dutch cohort scar size as well as IFN-γ production upon 7 day Mtb stimulation of PBMCs, and for the Indian cohort the frequency of IFN-γ producing CD4[+] T-cells following 12 h BCG stimulation of whole blood, suggest no difference in vaccine take between the functional groups. Thus, despite the observed strong differences in functional capacity to control mycobacterial growth, classical parameters of vaccine-induced immunity did not differ and could not discriminate between the three groups. Further research is needed to investigate this lack of functional response in more detail and unravel its mechanistic basis.

There are a few additional considerations that should be taken into account when evaluating these data. Firstly, Dutch volunteers in this study were vaccinated with BCG-Bulgaria, the Indian revaccinees with BCG Russia, whereas most other studies reporting BCG-induced growth control used BCG Denmark for vaccination. We cannot therefore exclude a role for the different BCG strains in the differences between studies, although the data reported herein are in agreement with our previous work[18,38] in which we used BCG Denmark.

Moreover, our in vitro growth inhibition assay used BCG Pasteur, which lacks RD3 compared to the strains used for vaccination, albeit we do not believe the absence of 10 genes accounts for the differences observed. Secondly, the sample post-BCG vaccination for the Dutch cohort was collected at 3 months post-vaccination, which is later than in most other studies assessing functional growth control. Thirdly, participants of the Indian BCG-revaccination could choose to be revaccinated or not thereby introducing a selection bias. However the Dutch participants volunteered to take part in the BCG vaccination study, thus also recruiting individuals to receive a BCG vaccination, is potentially leading to a similar bias. Fourthly, groups were relatively small upon classification based on the functional capacity to control BCG growth, nevertheless interesting differences were identified. Fifthly, In this study we could not use the full potential of all accessible flow cytometry data as the different cohorts were measured on different analyzers either by conventional flow cytometers or spectral analyzers and with different marker panels. Despite overlapping markers, the choice of clones and fluorochromes was dependent on the analyzer available, minimizing options for direct comparisons. Sixthly, analysis strategies for scRNAseq can be debated, here we preferred Seurat single-cell methodology over EdgeR-QLF pseudobulk analysis. In a small subset of data we compared the methods and Seurat single-cell methodology did not overestimate the number of DEGs. Finally it would be very interesting to investigate the role of additional stimuli besides LPS on the single-cell RNA level, as well as a time point 12 months after BCG (re)vaccination in future work. Despite these limitations, the study provided important new insights, which are of significance e.g. in monitoring immunity and evaluation of novel candidate vaccines against TB. In the TB biomarker field, multiple groups have employed "naturally protected" individuals (also called resistors) to identify surrogates for protection that could be used in evaluation of vaccine-induced immunity. However, our current work qualifies the use of these biomarkers as surrogates of natural ("already controllers") protection may not depend on the same mechanisms as vaccine-induced protection such that caution should be taken in monitoring vaccine-induced immunity using markers from resistor cohorts.

In conclusion, we report that the capacity to control mycobacterial growth is already detectable prior to BCG vaccination in a significant minority of two widely different (Dutch and Indian) populations, but that control can also be induced in a sizable group of additional individuals upon BCG vaccination and revaccination. In around half of the individuals no control was observed either before or after BCG (re)vaccination. Importantly, the gene-expression profiles of these groups were very different and indicate activation of different pathways leading to mycobacteria growth control capacity. These data thus show that functional control of mycobacterial growth can be mediated by multiple pathways, and also show that vaccination (such

as BCG) induced control may differ from control present already prior to vaccination.

## Methods

### Ethical approval

The 300BCG study was approved by the Arnhem-Nijmegen Medical Ethical Committee with number NL58553.091.16. All studies were performed in accordance with the declaration of Helsinki.

The Indian BCG revaccination study was performed according to guidelines of the Helsinki Declaration and was approved by the Institutional Ethics Review Committee of St. John's Medical College Hospital, Bangalore, IEC Ref no: (IEC/1/896/2018).

### 300BCG study

The 300BCG cohort[24,39] included 325 healthy adult volunteers of Western European ancestry between April 2017 and June 2018 in the Radboud University Medical Center, Nijmegen, The Netherlands. Written informed consent was obtained before blood was collected and 0.1 mL BCG (BCG-Bulgaria, InterVax) was administered intradermally in the left upper arm. Three months after BCG vaccination, additional blood samples were collected. Participants did not receive any prior BCG vaccination, nor any other vaccination in the 3 months preceding inclusion into this study, or during the study. Finally, participants that reported fever in the 4 weeks preceding study start were excluded from participation. BCG scarring was measured at 3 months after vaccination. In the present study we analyzed 42 participants of the original study in great detail, these being selected to contain 21 good and 21 poor responders to *Staphylococcus aureus* (SA) stimulation after BCG vaccination with IL-1β as biomarker of trained immunity[17].

All 42 participants had samples at baseline and at 3 months. Our subgroup comprised 20 women (average age 24.3 years (19.7–54.9 yrs)) and 22 men (average age 27.5 years (18.5–70.6 yrs)) identified by self-reporting and were thereby sex and age matched (source data file).

### Indian BCG-revaccination study

Healthcare workers aged 18–24 of St. John's Medical College Hospital, Bangalore, India[11] were invited to participate in the study from October 2019 to June 2021. All recruited individuals confirmed BCG vaccination at birth. Participants were screened for Mtb infection by the standard QFT TB Gold In-tube test (Qiagen) performed at Department of Microbiology, SJMCH, India, and 66 IGRA- subjects were enrolled for the study. Data were successfully obtained for 58 subjects. A prospective observational study was conducted to evaluate the effect of BCG-revaccination. Volunteers were given the choice of either being BCG revaccinated (BCG-RV, Group 1) or not (BCG-NRV, Group 2) after written informed consent was obtained. BCG vaccine (TUBERVAC™, Russian BCG strain manufactured at Serum Institute of India, Pune, India), used widely in the Indian national immunization program, was administered intradermally at day 0 at an adult dose of $2 \times 10^5$ to $8 \times 10^5$ CFU in participants in the BCG-RV ($n = 35$); BCG-NRV ($n = 31$) subjects who were not BCG revaccinated served as control. Blood was collected from participants at days 0 (T0), day 1 (T1), 10–12 weeks (T2), after BCG-revaccination.

### PBMC and whole blood stimulation and cytokine detection

Freshly isolated PBMCs from 300BCG participants were resuspended in Dutch modified RPMI 1640 medium (Roswell Park Memorial Institute, Invitrogen), supplemented with 50 μg/ml gentamicin (Centrafarm), 2 mM Glutamax (Gibco), and 1 mM pyruvate (Gibco). In round bottom 96-well plates (Greiner) $5 \times 10^5$ PBMCs were stimulated with RPMI (medium control), heat-killed *S.aureus* ($10^6$ CFU/ml), heat-killed

*M.tuberculosis* H37Rv (5 μg/ml), *C.albicans* ($10^6$ CFU/ml, strain UC820) or *E. coli* LPS (10 ng/ml, serotype 055:B5, Sigma-Aldrich) and incubated at 37 °C with 5% $CO_2$. Supernatants were collected after 24 h for innate cytokines and 7 days for IFN-γ and stored at −20 °C until further analysis. IL-1β levels were measured by ELISA (R&D Systems), and IFN-γ concentrations by luminex (ProcartaPlex, Thermo Fisher) according to the manufacturer's protocols.

Whole blood samples (400 μl) from the Indian BCG-revaccination cohort were stimulated for 12 h with BCG ($0.2 \times 10^6$ CFU/ml) in the presence of costimulatory antibodies anti-CD28/CD49d (0.5 μg/ml) at 37 °C. After 2 h of culture Brefeldin A and monensin (Biolegend) were added (1:1000 from stock). After stimulation cells were treated with 2 μM EDTA (Sigma), red blood cells were lysed with FACS Lysing solution (Becton Dickinson) and fixed white blood cells were cryopreserved. Intracellular staining was performed with CD3 (clone UCHT1), CD14 (clone M5E2) (Biolegend), CD4 (clone SK3), CD8 (clone RPAT8), IFN-γ (clone B27) and perm/wash reagents according to manufacturer's instructions (BD Biosciences). Cells were acquired on a BD Aria Fusion and analyzed using FlowJo v10 (BD Biosciences). Gating strategies are provided in Supplementary Fig. 4a.

### MGIA

We used the optimized 'in tube' MGIA Euripred protocol in our studies[18–20]. MGIA was run in three independent experiments for both the 300BCG cohort and the Indian cohort. Donors were randomly selected but pre and post-vaccination samples were always paired. Cryopreserved PBMCs were thawed and rested in RPMI (Gibco life sciences, Thermo Fisher Scientific Inc., Bleiswijk, the Netherlands) supplemented with glutamax (Gibco) and 10% FBS (Hyclone, Thermo Fisher Scientific Inc.) (=R10 medium) at a concentration of $2 \times 10^6$ cells/mL for 2 h in the presence of benzonase (20 U/ml, Merck, Amsterdam, the Netherlands). After resting, cells were washed with R10 medium and counted with a Casy Cellcounter (Roche, Woerden, the Netherlands). $1 \times 10^6$ PBMCs were co-cultured for 4 days in RPMI supplemented with glutamax and 10% autologous human serum with 2.38 logCFU (±0.17 SD) of the *M.bovis* BCG Pasteur (P3) strain on a rotator in a 37 °C humidified $CO_2$ incubator in a final volume of 600 μl. All samples were run in duplicates. After 4 days, 100 μL supernatant was harvested and stored for future analysis and the remaining 500 μl per sample were transferred to a PANTA/Enrichment supplemented MGIT tube (Becton Dickinson, Erembodegem, Belgium) and placed in a BACTEC MGIT 960 system (BD) until time to positivity (TTP) was reached. All tubes included in the analysis were checked visually for possible contamination. Samples reaching positivity within 100 h were considered contaminated and thus deleted as false positive, since the inoculum of 2.38 logCFU BCG is expected to reach TTP after more than 250 h. As a control for the BCG inoculum all experiments included a serial dilution ($10^7$–$10^2$) of the BCG stock for time to positivity in PANTA/Enrichment supplemented MGIT tubes and plating on Middlebrook 7H10 agar plates, supplemented with 10% OADC (BD) for CFU determination. When all dilutions showed colonies on the 7H10 plates, plates were scanned on a Canon Scanner 9000 F and colonies were counted using ImageJ software. CFUs were converted to logCFU and plotted against TTP. Linear regression analysis was applied (GraphPad Prism software v8.1), all samples were transposed and data plotted as logCFU.

### Measurement of cytokine production after BCG stimulation

A 40-plex chemo-cytokine Bio-plex assay (*Luminex®*) was performed according to manufacturer's instructions (Bio-Rad, Veenendaal, The Netherlands) on supernatants collected after 96 h of BCG stimulation (from the functional MGIA assay). Samples were acquired on a Luminex 200 system with Bio-Plex manager software (v6.1).

## Inflammatory proteome analysis of plasma by proximity extension assay

Data on circulating plasma concentrations of inflammatory markers[24] were re-assessed here according to the functional classification of the participants. Briefly, plasma inflammation markers were measured using the commercially available Olink Proteomics AB Inflammation Panel (92 inflammatory proteins), using a Proceek Multiplex proximity extension assay. Detected proteins were normalized according to inter-plate controls to minimize inter-assay variation and reported on a log2 scale as normalized protein expression values.

Heatmaps were created using hierarchical clustering on rows and columns with Euclidean distance and average linkage method using Morpheus software (Broad Institute).

## Flow cytometry

300BCG study: We assessed myeloid and lymphoid immune cell levels[40,41] by five different 10-color flow cytometry panels as shown in supplementary information table 2. We focused on a set of 31 manually annotated immune cell subpopulations out of 58 that could be identified using these panels. To minimize biological variability, cells were processed immediately after blood sampling and typically analyzed within 2–3 h after sample collection on a 10-color Navios flow cytometer (Beckman Coulter) equipped with three solid-state lasers (488 nm, 638 nm, and 405 nm). Cell populations were gated manually. Calibration of the machine was performed once a week, and little adjustment to the machine setting had to be made during the inclusion period of the study. Analysis was performed using OMIQ analysis software (www.omiq.ai). Gating strategies are provided in Supplementary Fig. 4b.

Indian BCG-revaccination study: Priority was given to the functional assay, but all samples with at least $1 \times 10^6$ cells remaining were also subjected to detailed phenotyping by spectral flow cytometry (Supplementary Table 1)[25]. In short, a minimum of $1 \times 10^6$ rested cells were incubated with cell viability dye in PBS, washed in PBS/0.1%BSA (Sigma, Merck Life Science NV, Amsterdam, the Netherlands) followed by blocking Fc receptors with 5% human serum in PBS for 10 min, cells were washed twice and subsequently stained for chemokine receptors for 30 min at 37 °C. After washing, cells were incubated with all other surface markers at 4 °C for an additional 30 min. Cells were washed twice and fixed for 10 min at RT with 1% paraformaldehyde (Pharmacy LUMC, Leiden, The Netherlands). Cells were washed once more and stored at 4 °C for acquisition on a 5 L Cytek®Aurora (Cytek Biosciences, Fremont, CA, USA) the following day at the LUMC flow core facility (https://www.lumc.nl/research/facilities/fcf/). Analysis was performed using OMIQ analysis software (www.omiq.ai) and volcano plots were generated with the integrated EdgeR tool. Gating strategies are provided in Supplementary Fig. 4c.

Datasets were further exported for univariate analysis and tested by Kruskal–Wallis with Dunn's multiple test correction. When comparing pre and post-BCG vaccination, samples were tested using the Wilcoxon test.

## scRNASeq, data processing and analysis

For all 42 individuals tested in MGIA, both at pre and post-vaccination samples were generated for scRNAseq from exactly the same vial of PBMCs. Samples were generated by stimulation of $5 \times 10^5$ rested PBMCs for 4 h with 10 ng/mL LPS (serotype 055: B5; Sigma) or left unstimulated at 37 °C, 5% $CO_2$. LPS was selected for restimulation as heterologous innate immune activator. Upon stimulation, cells were washed in PBS and pooled based on the individual donor, time point of vaccination, and stimulation condition. Pools contained amaximum of 5 samples. As the scRNAseq samples originated from the same vial as the MGIA was performed, the batches processed for sequencing related to the three independent experimental runs for the MGIA. Each MGIA run yielded 10–12 pools of scRNAseq samples. In total 168 samples were processed for single-cell RNAseq at the Leiden Genome Technology Center, Leiden University Medical Center, Leiden, The Netherlands (Supplementary Fig. 1). Briefly, single-cell gene-expression libraries were generated on the 10x Genomics Chromium platform using the Chromium Next GEM Single Cell 3' Library & Gel Bead Kit v3.1 and Chromium Next GEM Chip G Single Cell Kit (10x Genomics) according to the manufacturer's protocol. Libraries were sequenced on a NovaSeq 6000 S4 flow cell using v1 chemistry (Illumina) with 28 bp R1 and 90 bp R2 run settings.

## Data pre-processing and demultiplexing

In each library, bcf2fastq Conversion Software (Illumina) was used to convert BCL files to FASTQ files, along with sample sheet including 10x barcodes. The proprietary 10x Genomics STAR in CellRanger pipeline (v3.1.0) was used to align read data to GRCh38/b38 (downloaded from 10X Genomics). We set the parameter of setting of expected cells to 2000. Finally, gene-expression matrix was generated which recorded UMIs count of each gene in each cell.

In each library, using pre-mapped bam files, duplicates were removed and cells were assigned to their individuals of origin using souporcell (v2.0)[42] and, 91.98% cells were retrieved for downstream analysis. Subsequently, the souporcell tool was employed to cluster cells based on allele counts using hierarchical clustering strategy and assign individuals to clusters. Further, genotype dataset of each individual was used to cross-check the consistency between the assigned individuals via souporcell and individual phenotypes id.

## Data quality control

Cells with (i) MT-genes percentage more than 25% and (ii) number of detected genes less than 100 or more than 5000 were removed. Also, only genes expressed in at least 5 cells (leaving ~200k cells and 21,975 genes) were considered for the downstream analysis.

## Data integration and clustering

Seurat (v4.0.0)[43] package of R (v4.0.2) was used to integrate and analyze all data together. In brief, at first, for each independent dataset from each pool, UMI counts were normalized (log(10,000x + 1)) and the top 2000 variable features were selected using NormalizeData and FindVariableFeatures function with default parameters, respectively. Later, repeated features were identified across all independent datasets, and utilized for scaling and PCA analysis on each dataset. Instead of canonical correlation analysis, in order to speed up calculation for integration and avoid overcorrection[43], reciprocal PCA was used via the SelectIntegrationFeatures function to detect integration anchors. Followed by IntegrateData, the integrated dataset was scaled and clustered using default parameters. Cell clusters were further annotated combining the results from SingleR[44] package of R (HumanPrimaryCellAtlasData, BlueprintEncodeData, MonacoImmuneData, DatabaseImmuneCellExpressionData, and NovershternHematopoieticData were selected as reference) and the expression level of known cell markers (CD4 T-cells: IL7R, CD3D; CD8 T-cells: CD8A, CD8B; Monocytes: CD14, IL1B; NK cells: GNLY, NKG7; B-cells: CD79A; mDC: HLA-DPA1, HLA-DPB1; pDC: CTSC, TSPAN13; Platelet: PPBP). We also found a group of T-cells with a high expression of HSPA1A, HSPA1B and labeled as HSP(T). After removing undefined cells, the remaining cells (CD4 T-cells = 65,170, CD8 T-cells = 31,926, Monocytes = 31,601, NK cells = 28,796, B-cells = 21,356, mDC = 2211, pDC = 1223) across all individuals and conditions were used for downstream analysis.

**Differentially expressed genes tests and enrichment analysis**
Differential gene-expression analysis was performed using FindMarkers/FindAllMarkers function with Wilcoxon Rank Sum test (two-sided) in Seurat. Genes expressed in at least 10% of the cells and having p_val_adj <0.05 (BH correction) were considered significant.

For the enrichment test, significant gene sets were subjected to enrichGO and enrichKEGG function (pAdjustMethod = "BH", qvalueCutoff = 0.05) of ClusterProfiler (v3.18.1)[45] package of R, separately. Pathways with BH correction (FDR) < 0.05 were regarded as significant.

Analysis of pathways differentially expressed between acquired and already controlling samples was performed using pathway comparisons analysis in Ingenuity Pathway analysis (Qiagen).

**Statistics & reproducibility**
From the original 300BCG cohort of 325 individuals who received BCG vaccination a selection of 42 participants was included in this study. The selection was based on the response to *Staphylococcus aureus* (SA) stimulation after BCG vaccination with IL-1β as biomarker of trained immunity. Researchers were blinded to this information while conducting experiments. For the Indian validation cohort all available samples were included in the study. All experiments were performed in multiple runs in randomized groups but with pairing of the pre- and post-BCG vaccination samples (Supplementary Fig. 1). All data were analyzed assuming a non-Gaussian distribution, and therefore non-parametric testing was applied. No statistical method was used to predetermine sample size and no data were excluded from the analysis. Statistical analysis was performed using GraphPad Prism software v8.1 and v9.3.1, OMIQ software, and R packages as stated in the corresponding methods section.

**Reporting summary**
Further information on research design is available in the Nature Portfolio Reporting Summary linked to this article.

## Data availability
All data generated or analyzed during this study are included in this published article (and its supplementary information files in particular the source data file). Single-cell RNA sequencing data of the current study is deposited in the European Genome-Phenome Archive repository with ID: EGAS00001006990. Canonical pathways analysis identified the pathways from the QIAGEN Ingenuity Pathway Analysis library with the QIAGEN's Knowledge Base. Source data are provided with this paper.

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

## Acknowledgements

We would like to thank all volunteers from the 300BCG cohort for participation in the study. The authors thank the volunteers for participating in this study and acknowledge the contributions of clinical research workers at St John's Research Institute. We would like to thank Suzanne van Veen for assisting with statistical analysis. This work was supported by grants from the European Commission (EC) HORIZON2020 TBVAC2020 (Grant Agreement 643381)(KEvM, THMO & SAJ); ERC Advanced Grant (#833247) and Spinoza grant of the Netherlands Organization for Scientific Research (MGN); ERC starting Grant (948207), NWO ASPASIA and Radboud University Medical Centre Hypatia Grant (YL) and China Scholarship Council PhD scholarship (WL). The funders had no role in study design, data collection and analysis, decision to publish, or preparation of the manuscript.

## Author contributions

S.A.J., K.E.vM., and T.H.M.O. designed the study, analyzed data, and wrote the manuscript. K.E.vM. performed all experiments. W.L. and Y.L. analyzed the scRNAseq data. S.J.C.F.M.M., V.A.C.M.K., H.J.P.M.K., and L.A.B.J. were responsible for sample collection of the 300BCG cohort designed and supervised by M.G.N., who also acquired funding to perform the study. V.A.C.M.K. processed samples for plasma analysis and H.J.P.M.K. performed 300BCG flow cytometry analysis. A.V., A.A., S.R., and V.A. designed and collected the Indian BCG revaccination study.

## Competing interests

L.A.B.J. and M.G.N. are scientific founders of TTxD and Lemba. All other authors claim no conflict of interest.
