## [Peer Review File · Nature Communications]

BCG vaccination-induced acquired control of mycobacterial growth differs from growth control preexisting to BCG vaccinationREVIEWER COMMENTS

Reviewer #1 (Remarks to the Author):

NCOMMS-22-49715-T

Tuberculosis remains a major cause of morbidity and mortality globally despite the availability of the BCG vaccine for over a century. This manuscript investigates the factors the influence mycobacterial growth control, a critical question for the development and assessment on new vaccines to protect against TB. Building on previous work by the team which identified 'Natural control' in some individuals, this study compared immunophenotypes, BCG-specific immune responses and both resting and LPS-induced transcriptional responses between individuals with natural control of BCG in vitro growth, those who acquired control after primary BCG vaccination and those who failed to control even after BCG vaccination. The authors also made efforts to validate the immunophenotypes, BCG-specific immune response findings from the Dutch participants in the 300BCG study in participants from a BCG re-vaccination study in India. The findings of this study, that there are different mechanism of action involved in 'natural control' compared to 'acquired control' has important implications for future vaccine development and the identification of correlates of protection for BCG.

General comments:

Please ensure there is consistency of tense, particularly in the introduction

One major concern is that the current presentation of data does not enable critical assessment. For several key figures, individual data points (or summary data providing indication of variance) and DEG lists are not provided thus the validity of interpretation of the data cannot be determined.

Specific comments:

Introduction:

- The introduction is long and particularly paragraphs 2-5 should be markedly reduced.

Methods:

- Please provide further detail of selection of the 42 participants from the BCG 300 study, was there sex matching between the good and poor Staphylococcus aureus IL-1 β responders? Was there age matching?
- Also, please clarify if the Staphylococcus aureus (SA) stimulation IL-1 β responses relate to the BCG-vaccination induced changes (i.e. fold change post vaccination).
- For MGIA please state which strain of BCG is used.
- Pg 8 the following sentence is missing a word, please correct "Genes expressed in at 10% cells and having $p_val_adj < 0.05$ (BH correction) were considered significant."
- Representative gating strategies should be provided for the flow cytometry and spectral cytometry.

Results

- Please provide details of the ages of the participants from the 2 cohorts.
- Please remove figure 1C and the description of it in the results section. The information provided for it is already present and described for figure 1 A-B.
- Figure 1. Please correct participant numbers in figure legend B, the numbers for the left and middle panels appear to be swapped.
- "The capacity to control BCG outgrowth was not related to the original IL-1 β response to

Staphylococcus aureus (data not shown).” Given the IL-1 β response to Staphylococcus aureus response was used to select participants this result is important and the data should be provided as a supplementary figure.

- To avoid confusion, please use consistent terminology for the functional groups e.g. the already, acquired, and no control, groups are referred to as “natural control, BCG induced acquired control, lack of control” on pg 9.
- Figure 2D, please provide the pre (linked to post) data for the ‘no’ group as is done for the acquired group. Also please remove details of results from figure legend and report in manuscript text. (e.g. “17% of individuals”, “In the BCG revaccination group 46% (11/24) of individuals, that did not already control prior to vaccination, acquired the capacity to control BCG after vaccination.”, “The acquired and no control group consist of 55% and 62% women respectively.”, “Already pre vaccination was significantly different from acquired pre vaccination in a direct comparison using Mann-Whitney testing.”)
- The heatmaps presented in figures 2A, 2B, 2C and 2E do not provide any measure of variance and no statistical testing. Thus based on presentation of the medians alone it is difficult to determine the implications of this data. If retained, these figures should be supported by presentation of data that shows individual data points or the variance in individuals in the groups (i.e. boxplots as was done in figure 2F or tables).
- Figure 2E, please provide the pre data for the ‘no’ group.
- Figure 2F. Please provide explanation in figure legend or correct cell populations in grey font in volcano plots.
- When referring to the groups in the study from India (both main manuscript text and figure legend) please ensure the term “BCG-revaccination” is used at all times appropriate.
- “Interestingly, samples from individuals that acquired control by revaccination had the highest cytokine concentrations pre-revaccination.” It is not clear to me that the data/analysis supporting this statement was provided, please provide the data/analysis or amend the statement.
- Please correct term “flowcytometry” throughout the manuscript and figure legends
- “Control prior to BCG revaccination, was associated with increased frequencies of CD14+CD16+ intermediate monocytes and reduced classical CD14+CD16- monocytes. There was a trend towards increased non-classical monocytes (CD14-CD16+) in already controlling individuals.” Please specify the comparator groups for these statements, moreover, assuming the comparator is the acquired pre as per the volcano plot, the data does not support the statements made regarding differences in classical and non-classical monocytes. Please amend these statements.
- “Looking back in the 300BCG data, we observed a similar shift in the monocyte populations, in particular a decrease in classical monocytes in samples with control, but also increased frequencies of intermediate and non-classical monocyte subsets upon acquisition of control by BCG vaccination (Figure 2A).” The patterns of expression between the 300BCG and Indian cohort do not appear to be similar as there is no difference in these monocyte subsets between the acquired pre and acquired post data from the Indian participants. Also, given for the 300BCG participants median only is presented with no measure of variance, it is not clear how robust these apparent differences are.
- Samples were selected on the basis of Staphylococcus aureus IL1 β responses and grouped by BCG responses. Please provide further explanations as to why LPS stimulation was selected for the single cell analysis (as opposed to other stimuli for which BCG-trained immunity effects have been observed e.g. Staphylococcus aureus or Candida albicans). Given LPS was chosen as the stimulus, it would be of high relevance to provide the LPS cytokine responses for the subset of participants from the 300BCG cohort who were included in this study and the analysis as to whether the cytokine responses also tracked with BCG control groupings.

- For reporting of results of the single cell transcriptomic analysis, please provide the specific numbers in addition to or rather than descriptions e.g. “a high number of DEGs”, “similar DEG numbers”, “a large number of DEGs”.
- Currently there is no statement on data availability. At a minimum the full gene lists of DEGs should be made available.

Discussion

- “Unfortunately, no data are available that would be more classic read outs of vaccine uptake or vaccine induced adaptive immunity such as antigen specific Elispots or cytokine secretion in response to stimulation (e.g. intracellular cytokine measurement by flowcytometry) 42. It would be highly valuable to include classical parameters of vaccine take in future studies.” Given PBMC are available from the participants in these studies, why were such “highly valuable” measures not done.
- The fact that participants Indian BCG revaccination study participants could chose to be BCG re-vaccinated (BCG-RV, Group 1) or not (BCG- NRV, Group 2) should be listed as a limitation of the study.

Reviewer #2 (Remarks to the Author):

This interesting manuscript uses the mycobacterial growth inhibition assay, a functional assay shown to correlate with in vivo protection in animal models, to define 3 groups of BCG vaccine recipients – those who have baseline control (which does not change post BCG, called natural immunity here), those who’s control of mycobacterial growth improves post BCG vaccination, and those who’s control does not change post BCG. The authors then looked at single cell transcriptomic data by group and found differences in gene expression profile across these 3 groups. This paper has some interesting data in it and the finding that ‘natural’ ability to control mycobacterial growth in vitro and BCG induced control may differ in mechanism is novel and of relevance to the field of TB vaccine development.

Specific comments:

1. I think it is overstating the literature to say in the abstract that BCG induces considerable non specific effects. It induces some -but this is not equivalent to the >80% protection seen against TBM in infancy and childhood.
2. I am surprised the authors do not reference a recent paper in the introduction which provides some of the best quality evidence for non specific effects – Prentice et al, 2021.
3. The authors use the term ‘outgrowth’ – whereas all the MGIA assay is doing is quantifying live BCG present – so growth. I’m not clear why they think this should be called outgrowth.
4. In the volunteers who have good baseline control of mycobacterial growth, the authors have excluded (as well as they can) recent M.tb exposure but what about NTM exposure. Given the huge number of cross reactive antigens, NTM exposure may contribute to the baseline control observed in this study.
5. The last two sentences of the introduction don’t entirely make sense. The control present at baseline is in BCG naïve subjects- so it is incorrect to say these data provide evidence that BCG can induce diverse pathways. These data provide evidence that diverse pathways to control mycobacterial growth are possible – but they are only showing one pathway post BCG vaccination. The sentence at the end of the discussion is more accurate.
6. The 42 subjects selected from the larger parent study were selected on the basis of control of Staph aureus growth in vitro. But the study here looks at control of M.tb. There is no correlation between control of Staph and control of M.tb – which weakens the justification for selection criteria in this relatively small sample size.

7. How many of the 42 subjects were from the Dutch study and how many were from the Indian study? They are not comparable at all given different prevalence of both M.tb and NTM exposures.
8. Reference 19 refers to the optimised EURIPRED protocol for the MGIA, not reference 17 (M&M, MGIA).
9. Given the flow cytometry was conducted on fresh cells, this must mean the Dutch and Indian study had flow cytometry conducted in different labs and on different machines. Can the authors confirm if this is correct and what steps were taken to standardise. Or were the Indian samples processed from cryopreserved PBMC? This is unclear – either way the data may not be directly comparable.
10. In the natural controllers, did the authors assess baseline central memory responses to mycobacteria by proliferation or cultured elispot?
11. It would have been really interesting to also look at a later time point perhaps 12/12 post BCG to see how durable these effects and these differences are. Many of the non specific effects demonstrated after BCG vaccination are present at 3 but not 12 months (e.g. Kleinnijenhuis 2012).
12. The lack of any assay to measure vaccine ‘take’ is a limitation of this work, as the authors admit in the discussion – as they cannot exclude lack of ‘take’ in those who do not respond to BCG in the MGIA. Are there cells left for them to run an ELISPOT assay (more sensitive than flow) to include this data?
13. The authors comment that the Dutch cohort received BCG Bulgaria – but it is also noteworthy that the Indian cohort received BCG Russia.
14. Comparing Figure 1A and 2D the range and spread of baseline control is very different between the Dutch and Indian cohort. Were these samples run at the same time?

Reviewer #3 (Remarks to the Author):

This manuscript addresses differences in mycobacterial growth inhibition capacity in healthy Dutch adults who have been vaccinated with BCG. There is considerable variation in mycobacterial growth inhibition capacity on a population level, and this study utilizes single-cell RNA-seq to identify potential mechanisms and pathways that underlie differences between mycobacterial growth inhibition capacity. A mycobacterial growth inhibition assay (MGIA) utilizing cryopreserved PBMCs incubated with BCG was used to evaluate mycobacterial growth inhibition capacity and divide participants into categories of pre-existing (natural) control, BCG-acquired control, or no control. The main conclusion is that mycobacterial growth inhibition naturally present in healthy adults prior to BCG vaccination employs different mechanistic pathways than mycobacterial growth inhibition that is acquired following BCG vaccination, thus suggesting that mycobacterial growth inhibition can be achieved by diverse mechanisms. The manuscript lacks focus and many of the figures are illegible as currently presented, thus making it difficult to draw meaningful conclusions from the study.

Comments:

- Participants reported in this manuscript were selected from a larger study of 325 healthy adults vaccinated with BCG. The authors selected 42 participants (21 good and 21 poor responders to *Staphylococcus aureus* (SA) stimulation as a marker of trained immunity) for further analysis in this manuscript. More demographic information (age, race/ethnicity) should be provided for the 42 participants selected. Given that the participants were selected based on induction of trained immunity, the data reported in the manuscript will need to be interpreted with caution as there is a selection bias in participants and the results may not be

applicable broadly to BCG-vaccinated individuals.

- Individuals with 'acquired' mycobacterial growth inhibition are defined as those with a change of ≥ 0.17 logCFU before and after BCG vaccination, although the rationale for this definition is not clear.
- Figure 2 is difficult to follow. Immune cell frequencies, plasma cytokine/chemokine markers, and cytokine/chemokines produced after in vitro BCG stimulation are presented in heat maps, each clustered by analyte and by participants that have been stratified into 6 groups. The authors observe 'branching' in the heat maps although the biological relevance is not clear. Reasons for reproducing sections of the heat maps in Figure 2B and 2C are not clear.
- The authors use an independent cohort of BCG re-vaccinated young adults (18-20yrs) in India to validate findings from their cohort of healthy Dutch adults. There are substantial differences between the Indian and Dutch cohorts, thus making comparisons and interpretation of the data between these two cohorts difficult. It is also not clear what value the immune profiling data from the Indian cohort add to the overall conclusions of the study, which are largely based on scRNA-seq data conducted only in the Dutch cohort.
- Immune profiling of samples from the Indian cohort are referenced in a submitted manuscript, and also presented in this manuscript, this creating the impression of duplication of data reported in the two manuscripts.
- The data presentation in Figures 3-5 are not clear, with the text not legible as currently presented.
- ScRNA-seq was conducted on PBMCs that were either incubated in media alone or stimulated for 4 hours with LPS. The rationale for analysis of gene expression in LPS-stimulated PBMCs is not clear; it is also unclear how gene expression in LPS-stimulated PBMCs provides further insight into mechanisms by which PBMCs restrict BCG growth.
- The manuscript oscillates between focusing on trained immunity and mycobacterial growth inhibition, two themes that are not necessarily directly linked, thus detracting from clarity in the overall message of the paper.
- The manuscript would benefit from revisions to reduce the text and make the manuscript concise with a more focused message. The figure legends are lengthy and difficult to follow. The figure legends include description and interpretation of the data, which should be in the Results section of the text rather than repeated in the figure legends.

Appendix 2: point-to-point reply to reviewer comments

Note: line numbers in the point-to-point reply refer to the manuscript version including track changes to facilitate easy retrieval of changes

REVIEWER COMMENTS

Reviewer #1 (Remarks to the Author):

General comments:

Please ensure there is consistency of tense, particularly in the introduction. One major concern is that the current presentation of data does not enable critical assessment. For several key figures, individual data points (or summary data providing indication of variance) and DEG lists are not provided thus the validity of interpretation of the data cannot be determined.

We thank the reviewer for the appreciation of the findings of this study and for the critical reading and comments raised. We have carefully looked into the consistency of tense throughout the whole manuscript and adjusted accordingly. We have now made all data available in a source data file, that includes all raw data used to generate the figures, including all DEGs with fold changes and p-values data. Furthermore the single cell RNA sequencing data have been deposited in the European Genome-Phenome Archive (ID:EGAS00001006990) and is accessible from there. The source data file also contains all individual data points for the soluble marker profiling as well as all flowcytometry results. We trust this satisfies the reviewer's request.

Specific comments:

Introduction:

- The introduction is long and particularly paragraphs 2-5 should be markedly reduced. *We have condensed the introduction and focused it more on the work relevant to the data presented here.*

Methods:

- Please provide further detail of selection of the 42 participants from the BCG 300 study, was there sex matching between the good and poor *Staphylococcus aureus* IL-1 β responders? Was there age matching?

*Participants were selected on their responsiveness of IL-1 β to heat-killed *S.aureus* post BCG vaccination as this is thought to be one of the key biomarkers for trained immunity. We have now included details on the IL-1 β responses of the participants to *S.aureus*, *C.albicans*, *M.tuberculosis* and LPS in supplementary figure 1. In panel a. we show the responses against these stimuli for the groups based on poor and good responses to *S.aureus* as was the initial classification. In panel b. we show the same data set but now split according to the three functional groups identified in the MGIA assay. We have also described this in the text (line 369-380, in version with track changes indicated). All individual data points for the IL-1 β data have also been included in the source file.*

Participants were sex-and age matched as can be seen in the table and figure below, this data have also been included in the source data. We also changed the text to "Our subgroup comprised 20 women and 22 men identified by self-reporting of sex and were age matched" (line 173-175).

S.aureus responder	Female	Male
good	11	10
poor	9	12

- Also, please clarify if the Staphylococcus aureus (SA) stimulation IL-1 β responses relate to the BCG vaccination induced changes (i.e. fold change post vaccination).

The original definition of good and poor responders was actually based on the difference in IL-1 β production before and after vaccination, the fold change of S.aureus induced IL-1 β production, as a biomarker of trained immunity. In supplementary figure 1a we now show the IL-1 β responses before and after BCG vaccination as log₂ fold change compared to the unstimulated (medium condition). IL-1 β concentrations are increased after BCG vaccination in the S.aureus good responders. Although the poor S.aureus responders already had a response to S.aureus prior to BCG vaccination, this response decreased upon vaccination. In this new supplementary figure we also grouped the participants according to the functional assay (MGIA) and have plotted S.aureus induced IL-1 β production for these groups. There was no association between the selection on S.aureus induced IL-1 β responses and the level of MGIA control, therefore we continued our analyses with the functional classification.

We have now clarified this in the text at lines 170-172. “In the present study we analyzed 42 participants of the original study in great detail, these being selected to contain 21 good and 21 poor responders to Staphylococcus aureus (SA) stimulation after BCG vaccination with IL-1 β as biomarker of trained immunity¹⁶.”

- For MGIA please state which strain of BCG is used.

We have now changed this in the text to “with 2.38 logCFU of the M.bovis BCG Pasteur (P3) strain” (line 226).

- Pg 8 the following sentence is missing a word, please correct “Genes expressed in at 10% cells and having p_val_adj < 0.05 (BH correction) were considered significant.”

We thank the reviewer for the noticing these missing words and have now corrected the text to “at least 10% of the cells” (line 333).

- Representative gating strategies should be provided for the flow cytometry and spectral cytometry. *We have added a new figure, supplementary figure 3, which shows all gating strategies for the flow cytometry used in this study and referred to them in the text in the Materials and Methods and in the results section (lines 216, 266, 280, 388,1004).*

Results

- Please provide details of the ages of the participants from the 2 cohorts.

We provide now all information related to age in the source data file and have included details in the cohort descriptions. For the Dutch cohort we adapted the text to “Our subgroup comprised 20 women (average age 24.3 years (19.7 – 54.9 yrs)) and 22 men (average age 27.5 years (18.5 – 70.6 yrs)) identified by self-reporting and were thereby sex and age matched” (lines 173-175). The age distribution was also plotted in response to comment 1 in the methods and shows a similar distribution.

For the Indian revaccination cohort we adjusted the text in the results section to “The primary BCG vaccination was administered directly at birth and considering the age of the participants (average age 20.2 years with a range of 18-24 years),” (lines 436-438).

- Please remove figure 1C and the description of it in the results section. The information provided for it is already present and described for figure 1 A-B.

As suggested by the reviewer, we have now removed figure 1C and adapted the text throughout the manuscript.

- Figure 1. Please correct participant numbers in figure legend B, the numbers for the left and middle panels appear to be swapped.

We thank the reviewer for spotting this error and have corrected the numbers of the participants in the figure legend.

- “The capacity to control BCG outgrowth was not related to the original IL-1 β response to Staphylococcus aureus (data not shown).” Given the IL-1 β response to Staphylococcus aureus response was used to select participants this result is important and the data should be provided as a supplementary figure.

We fully agree with the reviewer on this valid comment, requesting to show these characteristics of the participants included in the study. We have now added supplementary figure 1 that shows the IL-1 β responses of the participants in relation to S.aureus, C.albicans, M.tuberculosis and LPS, grouped according to the original S.aureus selection criteria (panel a) and in relation to the functional MGIA results (panel b), showing no correlation between mycobacterial growth control and S.aureus induced IL-1 β production. We also added a description of these data in the results section of the manuscript (lines 369-380).

- To avoid confusion, please use consistent terminology for the functional groups e.g. the already, acquired, and no control, groups are referred to as “natural control, BCG induced acquired control, lack of control” on pg 9.

We have used consistent nomenclature throughout the manuscript and changed natural control to already controlling.

- Figure 2D, please provide the pre (linked to post) data for the ‘no’ group as is done for the acquired group. Also please remove details of results from figure legend and report in manuscript text. (e.g. “17% of individuals”, “In the BCG revaccination group 46% (11/24) of individuals, that did not already control prior to vaccination, acquired the capacity to control BCG after vaccination.”, “The acquired and no control group consist of 55% and 62% women respectively.”, “Already pre vaccination was significantly different from acquired pre vaccination in a direct comparison using Mann-Whitney testing.”)

We agree with the reviewer that the information in this legend is too detailed and have corrected this in the new figure legend. We also have added the pre (linked to post) data for the no -control group in the same way as we have shown this for the acquired group and adjusted the legend and text accordingly.

- The heatmaps presented in figures 2A, 2B, 2C and 2E do not provide any measure of variance and no statistical testing. Thus based on presentation of the medians alone it is difficult to determine the

implications of this data. If retained, these figures should be supported by presentation of data that shows individual data points or the variance in individuals in the groups (i.e. boxplots as was done in figure 2F or tables).

We thank the reviewer for these suggestions that help to improve the interpretation and quality of the data shown. We have now performed statistical analyses for all data sets included in the heatmaps for both the within group (pre-post vaccination to identify vaccine induced changes) as well as the between groups comparisons (differences associated with functional capacity). The statistical analyses are also supplied in the source data file.

For each heatmap we now visualize the significant p-values for the within group comparisons (Mann-Whitney matched-pairs with Benjamini-Hochberg multiple test correction) and the between groups comparisons for both the pre and post vaccination timepoints (Mann-Whitney U test with Benjamini-Hochberg multiple test correction) in supplementary figure 4a-d. Although we have performed multiple test correction on the statistics, we only display the FDR corrected p-values for figure 2c in supplementary figure 4c. Supplementary Figure 4a,b and d show the significance with uncorrected p-values, as these broad discovery datasets encounter an imbalanced large FDR factor in the statistical analysis and thereby underestimating the relevance of the data displayed.

In addition, we have shown all analytes that were discussed in the actual text in more detail now also as univariate plots in supplementary figure 4e and f as suggested by the reviewer.

- Figure 2E, please provide the pre data for the 'no' group.

We have now included this group in the heatmap and have made all individuals' data points available in the source data file.

- Figure 2F. Please provide explanation in figure legend or correct cell populations in grey font in volcano plots.

We have adjusted the font color in the volcano plots.

- When referring to the groups in the study from India (both main manuscript text and figure legend) please ensure the term "BCG-revaccination" is used at all times appropriate.

We carefully checked and adjusted when appropriate.

- "Interestingly, samples from individuals that acquired control by revaccination had the highest cytokine concentrations pre-revaccination." It is not clear to me that the data/analysis supporting this statement was provided, please provide the data/analysis or amend the statement.

We have amended the statement in the text.

- Please correct term "flowcytometry" throughout the manuscript and figure legends.

We have corrected the term throughout the manuscript and thank the reviewer for noticing.

- "Control prior to BCG revaccination, was associated with increased frequencies of CD14+CD16+ intermediate monocytes and reduced classical CD14+CD16- monocytes. There was a trend towards increased non-classical monocytes (CD14-CD16+) in already controlling individuals." Please specify the comparator groups for these statements, moreover, assuming the comparator is the acquired pre as per the volcano plot, the data does not support the statements made regarding differences in classical and non-classical monocytes. Please amend these statements.

We apologize that the description of the monocyte data was confusing. We have adapted the text and include reference to the comparator we refer to: "The already controllers displayed increased frequencies of CD14⁺CD16⁺ intermediate monocytes in comparison with individuals that acquired control by revaccination. Increased classical CD14⁺CD16⁻ monocytes prior to revaccination was present in individuals that were not able to control BCG growth even after revaccination. There was a trend towards increased non-classical monocytes (CD14⁻CD16⁺) in already controllers, compatible with their previously described role in BCG growth control." (lines 455-471).

- *"Looking back in the 300BCG data, we observed a similar shift in the monocyte populations, in particular a decrease in classical monocytes in samples with control, but also increased frequencies of intermediate and non-classical monocyte subsets upon acquisition of control by BCG vaccination (Figure 2A)." The patterns of expression between the 300BCG and Indian cohort do not appear to be similar as there is no difference in these monocyte subsets between the acquired pre and acquired post data from the Indian participants. Also, given for the 300BCG participants median only is presented with no measure of variance, it is not clear how robust these apparent differences are. We agree with the reviewer that the data representation did not permit full appreciation of the data. We have now included the statistical analysis of all phenotyping data in supplementary figure 4a. Since the only significant difference in monocyte populations was observed between pre and post vaccination samples that lacked control, we have modified the text and removed the suggestion that shifts in monocyte populations were common to all subgroups in the 300BCG cohort.*

- *Samples were selected on the basis of Staphylococcus aureus IL1b responses and grouped by BCG responses. Please provide further explanations as to why LPS stimulation was selected for the single cell analysis (as opposed to other stimuli for which BCG-trained immunity effects have been observed e.g. Staphylococcus aureus or Candida albicans). Given LPS was chosen as the stimulus, it would be of high relevance to provide the LPS cytokine responses for the subset of participants from the 300BCG cohort who were included in this study and the analysis as to whether the cytokine responses also tracked with BCG control groupings.*

We thank the reviewer for this critical, but valid, comment and would like to explain the rationale for choosing LPS in more detail. First of all, we wanted to stimulate predominantly the myeloid compartment to assess monocyte reprogramming, and not reactivate antigen specific memory T- or B-cells (which might take place when mycobacterial stimuli would be used). Secondly, we wanted to specifically assess responses to a heterologous stimulation compared to the in vivo BCG stimulation. Thirdly, pilot experiments included both LPS and S.aureus stimulation for 4 hours and in these LPS induced a greater number of DEGs than S.aureus compared to the unstimulated comparator.

Since LPS was actually included in the in vitro stimulation assay and IL-1β production measured, the data have now been added to the source data and are included in supplementary Figures 1a and b. The LPS induced IL-1β response post vaccination was significantly higher than at the pre vaccination timepoint for the S.aureus responders, but not for the non-responders. When grouping donors according to the functional MGIA results, we did not observe differences in LPS or S. aureus induced IL-1β between the groups prior to vaccination.

Thus based on the research question, pilot data and LPS induced cytokine profiles we decided to use LPS for restimulation for scRNA expression analysis. However, we do agree that additional stimuli would be very interesting and should be considered for future studies.

- For reporting of results of the single cell transcriptomic analysis, please provide the specific numbers in addition to or rather than descriptions e.g. “a high number of DEGs”, “similar DEG numbers”, “a large number of DEGs”.

We thank the reviewer for this comment and have added all numbers of DEGs to the text. In addition all DEG lists have been included in the source data file.

- Currently there is no statement on data availability. At a minimum the full gene lists of DEGs should be made available.

The reviewer is correct in this comment and all data shown in this manuscript, including DEG lists, are now available in the source data file according to the journal’s requirements. Raw data from the scRNAseq transcriptomic analysis is deposited at the European Genome-Phenome Archive (EGA), ID: EGAS00001006990.

Discussion

- “Unfortunately, no data are available that would be more classic read outs of vaccine uptake or vaccine induced adaptive immunity such as antigen specific Elispots or cytokine secretion in response to stimulation (e.g. intracellular cytokine measurement by flowcytometry) 42. It would be highly valuable to include classical parameters of vaccine take in future studies.” Given PBMC are available from the participants in these studies, why were such “highly valuable” measures not done. *We highly appreciate the reviewer’s comment to this matter and looked into all data available on both cohorts to asses vaccine take in both studies (there were no PBMCs available to perform Elispots to prove this point).*

*For the 300BCG Dutch cohort BCG scarring data was available, showing scars for all participants except one, and no differences in scar sizes between the MGIA classified groups. In addition, we were able to analyze the IFN- γ responses for all participants pre and post vaccination in a 7 day stimulation assay with heat-killed *M.tuberculosis* as stimulus. The data showed a significant difference when comparing the post vaccination to the pre vaccination timepoint and are now provided in supplementary Figure 2a-d. The IFN- γ responses to *M.tuberculosis* were plotted according to both the original group classification and according to the BCG-control phenotype grouping. Importantly, the BCG-no control group showed a significant increase in the IFN- γ response to heat killed *M.tuberculosis* stimulation, suggesting vaccine induced (adaptive) immune recognition in this group. As mentioned this new data is now included in supplementary Figure 2 (a-d). Collectively we strongly believe that the equal scarring, as well as the increased post vaccination IFN- γ production in response to heat killed *M.tuberculosis* stimulation support equal vaccine take in all groups.*

For the Indian BCG-revaccination cohort, the induction of adaptive immune responses has been analyzed in another sub-study [BCG revaccination of young Indian adults enhances pro-inflammatory markers of trained immunity with concomitant induction of major anti-inflammatory pathways; Asma Ahmed, Himanshu Tripathi, K.E. van Meijgaarden, Simone A Joosten, Nirutha Chetan Kumar, Vasista Adiga, Srabanti Rakshit, Chaitra Parthiban, Pratibha Dwarkanath, Sudarshan Shivalingaiah, Srishti Rao, George D’Souza, Mary Dias, Tom H.M Ottenhoff, Mihai G. Netea, Annapurna Vyakarnam, submitted]. Ahmed et al performed a whole blood assay with 12 hour stimulation with BCG, followed by intracellular IFN- γ staining of CD4⁺ T-cells by flow cytometry. We show in supplementary figure 2 e. and f. that there was a significant increase in the response post vaccination in all subgroups, and which

was therefore unrelated to the BCG-control grouping. Taken together we believe these new results to have answered the important vaccine take question and have adjusted the results and discussion sections accordingly (lines 369-380, 439-442 and 775-782).

- The fact that participants Indian BCG revaccination study participants could chose to be BCG re-vaccinated (BCG-RV, Group 1) or not (BCG- NRV, Group 2) should be listed as a limitation of the study. *We agree with the reviewer and have added this to the discussion (line 791-792).*

Reviewer #2 (Remarks to the Author):

This interesting manuscript uses the mycobacterial growth inhibition assay, a functional assay shown to correlate with in vivo protection in animal models, to define 3 groups of BCG vaccine recipients – those who have baseline control (which does not change post BCG, called natural immunity here), those who’s control of mycobacterial growth improves post BCG vaccination, and those who’s control does not change post BCG. The authors then looked at single cell transcriptomic data by group and found differences in gene expression profile across these 3 groups. This paper has some interesting data in it and the finding that ‘natural’ ability to control mycobacterial growth in vitro and BCG induced control may differ in mechanism is novel and of relevance to the field of TB vaccine development.

We thank the reviewer for the critical reading of the manuscript and the appreciation of the relevance to the TB vaccine field. Questions raised by the reviewer are answered below in the point-by-point reply.

Specific comments:

1. I think it is overstating the literature to say in the abstract that BCG induces considerable non specific effects. It induces some -but this is not equivalent to the >80% protection seen against TBM in infancy and childhood.

We have adjusted the abstract stating “Bacillus Calmette-Guèrin (BCG)-vaccination induces not only protection in infants and young children against severe forms of tuberculosis (TB), but also against non-TB related all-cause mortality”.(lines 40-41, in version with track changes indicated).

2. I am surprised the authors do not reference a recent paper in the introduction which provides some of the best quality evidence for non specific effects – Prentice et al, 2021.

We apologize for omitting to cite this important reference and have now included it into the introduction of the manuscript.

3. The authors use the term ‘outgrowth’ – whereas all the MGIA assay is doing is quantifying live BCG present – so growth. I’m not clear why they think this should be called outgrowth.

We thank the reviewer for this comment and have adapted this accordingly throughout the manuscript.

4. In the volunteers who have good baseline control of mycobacterial growth, the authors have excluded (as well as they can) recent M.tb exposure but what about NTM exposure. Given the huge

number of cross reactive antigens, NTM exposure may contribute to the baseline control observed in this study.

We thank the reviewer for raising this always complicating factor while studying TB vaccines. We have addressed this in the manuscript but now emphasized this stronger in the discussion (line 738-744).

“Already control of mycobacterial growth was observed in approximately a quarter of Dutch donors, which is an unexpectedly high proportion in this very low burden country. Already control was also observed in 17% of Indian donors, who were nevertheless vaccinated a birth. We have previously described that recent exposure to patients with contagious active pulmonary TB resulted in a high frequency of individuals with growth control capacity¹⁸. However, with an annual TB incidence of ~4.7 per 100.000, mostly amongst foreign-born immigrants, it is highly unlikely that 25% of the Dutch population that was recruited into the study was recently exposed to a case of active pulmonary TB.”
And lines 752-761, “Alternative to exposure to Mtb, exposure to other, less pathogenic, non-tuberculous mycobacteria (NTM) such as M. avium, M. kansasii or M. marinum might induce similar responses, although this is as yet speculative. Of note, the proportion of controllers prior to vaccination in this study is similar to what we previously observed in a cohort of healthy Dutch non-BCG vaccinated blood bank donors, where ~30% of healthy donors were able to control BCG growth¹⁸. The occurrence of similar frequencies (25-30%) of non-BCG vaccinated healthy individuals that control BCG in two independent cohorts in a low TB endemic country suggests that there are also other factors that may activate the state of control, or that some individuals have the already controlling capacity as a result of e.g. genetic variation or (recent) exposure to other pathogens, including NTM, but these data are unavailable for the current cohorts.”

5. The last two sentences of the introduction don't entirely make sense. The control present at baseline is in BCG naïve subjects- so it is incorrect to say these data provide evidence that BCG can induce diverse pathways. These data provide evidence that diverse pathways to control mycobacterial growth are possible – but they are only showing one pathway post BCG vaccination. The sentence at the end of the discussion is more accurate.

We agree with the reviewer and have changed the last sentences of the introduction to “This study shows that individuals with acquired growth control upon BCG vaccination, express different functional pathways in comparison with individuals that already control prior to vaccination.” (lines 136-138).

6. The 42 subjects selected from the larger parent study were selected on the basis of control of Staph aureus growth in vitro. But the study here looks at control of M.tb. There is no correlation between control of Staph and control of M.tb – which weakens the justification for selection criteria in this relatively small sample size.

We thank the reviewer for the opportunity to clarify. The selection of the 42 participants was based on the IL-18 response to heat-killed S.aureus. We did not perform any functional assays to assess control of S. aureus. As included now in supplementary figure 1, individuals were selected based on their in vitro response, in production of IL-18, in response to 24 hours of heat-killed S. aureus stimulation. Data for this selection is now also provided in the source data file as well as in supplementary figure 1 where we show that the S.aureus response grouping is not equal to the BCG-control grouping.

7. How many of the 42 subjects were from the Dutch study and how many were from the Indian study? They are not comparable at all given different prevalence of both M.tb and NTM exposures.

We agree that the Dutch cohort cannot be directly compared to the Indian cohort given the different Mtb and NTM prevalences as stated by the reviewer and apologize if that suggestion inadvertently was made.

In this study, we have included 42 participants from the Dutch 300BCG cohort in which we were able to identify the differential BCG-control grouping as described in the manuscript and to perform in-depth scRNAseq analysis.

In addition, we used the Indian revaccination cohort consisting of 33 participants to validate our findings of the differential BCG-control grouping and associated immune profiles. Reassuringly, despite their widely different origins, major findings of the work was evident and replicated in both cohorts, lending credibility to the validity of the new findings with regard to BCG controlling and immune phenotypes.

8. Reference 19 refers to the optimised EURIPRED protocol for the MGIA, not reference 17 (M&M, MGIA).

We thank the reviewer for spotting this error and have corrected this accordingly.

9. Given the flow cytometry was conducted on fresh cells, this must mean the Dutch and Indian study had flow cytometry conducted in different labs and on different machines. Can the authors confirm if this is correct and what steps were taken to standardise. Or were the Indian samples processed from cryopreserved PBMC? This is unclear – either way the data may not be directly comparable.

Indeed, staining of the samples on the 300BCG cohort was done on freshly isolated cells at the flow facility of the Radboud University Medical Center, Nijmegen, the Netherlands. The Indian samples were cryopreserved and stained upon thawing and resting in Leiden, as mentioned in the Materials and Methods section. Both cohorts were stained with different panels but all were analyzed at Leiden University Medical Center using the OMIQ software and the same steps to clean the data and used the same gating strategies when feasible. However we have employed the same markers to identify major cell subsets, eg live CD3+ cells as T-cells.

In downstream analyses, subset data were only compared within each of the cohort, thus the 300BCG samples were compared at pre and post vaccination time points within individuals or between the already, acquired and no control groups within the 300BCG cohort. Similarly, we also analyzed differences in cell subset frequencies within the Indian cohort.

Since we only performed within cohort analyses, and defined subsets using the same markers, we assess relative changes of subsets within that same cohort rather than absolute frequencies and therefore we considered the technical differences acceptable for a validation cohort (albeit not perfect).

We have added supplementary Figure 3 showing all gating strategies for the different cohorts and panels used. In addition, all frequencies are included in the source data file.

10. In the natural controllers, did the authors assess baseline central memory responses to mycobacteria by proliferation or cultured elispot?

*We thank the reviewer for raising this point. We did not have PBMCs available to perform baseline Elispots. However, we were able to analyze and include the IFN- γ responses for all 300BCG participants pre and post vaccination in a 7 day stimulation assay (perhaps reflecting memory responses even better than a short-term Elispot assay) with heat-killed *M. tuberculosis* as stimulus.*

Specifically, at the pre-vaccination (baseline) time point, the already controlling individuals produced similar levels of IFN- γ compared to the acquired or non-controlling groups (supplementary Figure 2d).

For the Indian BCG-revaccination cohort, the induction of adaptive immune responses have been analyzed in another sub-study [BCG revaccination of young Indian adults enhances pro-inflammatory markers of trained immunity with concomitant induction of major anti-inflammatory pathways; Asma Ahmed, submitted]. Ahmed et al performed a whole blood assay with 12 hour stimulation with BCG, followed by intracellular IFN- γ staining in CD4⁺ T-cells by flow cytometry. We show in supplementary figure 2 e. and f. that there was a significant increase in the response post vaccination which was not related to the BCG-control grouping. Here, specifically the already controlling group was very small which limits interpretation of the data, but in any case there is no tendency towards more IFN- γ producing CD4⁺ T-cells in the already controlling group.

11. It would have been really interesting to also look at a later time point perhaps 12/12 post BCG to see how durable these effects and these differences are. Many of the non specific effects demonstrated after BCG vaccination are present at 3 but not 12 months (e.g. Kleinnijenhuis 2012). *We do agree with the reviewer that this would have been a valuable addition but unfortunately these 12 months samples were not collected in this cohort and we addressed this comment in the discussion (lines 792-796).*

12. The lack of any assay to measure vaccine 'take' is a limitation of this work, as the authors admit in the discussion – as they cannot exclude lack of 'take' in those who do not respond to BCG in the MGIA. Are there cells left for them to run an ELISPOT assay (more sensitive than flow) to include this data? *We appreciate the reviewers comment to this matter and looked into all data available on both cohorts to see if there is a proxy for vaccine take in both studies, as there were no more PBMCs available to perform Elispots to prove this point. We have answered this question in detail also in our reply to reviewer 1 and included scarring and IFN- γ production upon 7 day M.tb lysate stimulation as proxies for vaccine take for the 300BCG Dutch cohort. For the Indian cohort we have included 12 hour BCG stimulated whole blood IFN- γ producing CD4⁺ T-cells as proxy. This data is now included in the supplementary Figure 2 and in the source data file.*

13. The authors comment that the Dutch cohort received BCG Bulgaria – but it is also noteworthy that the Indian cohort received BCG Russia. *We thank the reviewer for the comment and have included this in the discussion section (lines 785-786).*

14. Comparing Figure 1A and 2D the range and spread of baseline control is very different between the Dutch and Indian cohort. Were these samples run at the same time? *Indeed there is a different baseline in both data sets as they were not measured at the same time. In addition to this possible technical variation, we would like to emphasize that the Indian BCG revaccination participants all had had a priming BCG vaccination at birth and were more likely to have been exposed to M.tb or NTM prior to inclusion into the study. We have now included the information on the standard curves on both cohorts in the source data file.*

Reviewer #3 (Remarks to the Author):

This manuscript addresses differences in mycobacterial growth inhibition capacity in healthy Dutch adults who have been vaccinated with BCG. There is considerable variation in mycobacterial growth inhibition capacity on a population level, and this study utilizes single-cell RNA-seq to identify potential mechanisms and pathways that underlie differences between mycobacterial growth inhibition capacity. A mycobacterial growth inhibition assay (MGIA) utilizing cryopreserved PBMCs incubated with BCG was used to evaluate mycobacterial growth inhibition capacity and divide participants into categories of pre-existing (natural) control, BCG-acquired control, or no control. The main conclusion is that mycobacterial growth inhibition naturally present in healthy adults prior to BCG vaccination employs different mechanistic pathways than mycobacterial growth inhibition that is acquired following BCG vaccination, thus suggesting that mycobacterial growth inhibition can be achieved by diverse mechanisms. The manuscript lacks focus and many of the figures are illegible as currently presented, thus making it difficult to draw meaningful conclusions from the study.

We thank the reviewer for the comments. We have made sure that all figures now are of the highest quality possible but we do realize these pathway analysis figures are data dense. We further address the reviewers comments below in a point-by-point reply.

Comments:

- Participants reported in this manuscript were selected from a larger study of 325 healthy adults vaccinated with BCG. The authors selected 42 participants (21 good and 21 poor responders to *Staphylococcus aureus* (SA) stimulation as a marker of trained immunity) for further analysis in this manuscript. More demographic information (age, race/ethnicity) should be provided for the 42 participants selected. Given that the participants were selected based on induction of trained immunity, the data reported in the manuscript will need to be interpreted with caution as there is a selection bias in participants and the results may not be applicable broadly to BCG-vaccinated individuals.

*We thank the reviewer for this comment. We have included age to the demographic data available to the source data file and did not find any differences between the good or poor responders. All participants are of Western European origin as this was a inclusion criterium of the study, this is stated in the materials and methods section (line 162, in version with track changes indicated). The initial selection was based on the *S.aureus* induced IL-1 β responses, we have now added data to the data source file and manuscript showing that this selection criteria was not relevant for the breakdown into the BCG-control grouping (supplementary Figure 1). In supplementary Figure 1a we show all IL-1 β responses, pre and post BCG vaccination, to the heterologous stimuli of heat-killed *S.aureus*, *C.albicans*, *M.tuberculosis* and LPS for the original *S.aureus* grouping. In supplementary Figure 1b we show these same IL-1 β responses for the BCG-control grouping where we do not see a correlation between the *S.aureus* responses and the BCG-control grouping. Although we do not intend to state that the pathways identified in this study for BCG vaccination are applicable to all populations, we did use an Indian BCG revaccination study as validation cohort besides the Dutch cohort that showed the same BCG-control grouping. Further studies including single cell RNA sequencing for this and future cohorts should be performed in combination with the functional MGIA to confirm our findings.*

- Individuals with ‘acquired’ mycobacterial growth inhibition are defined as those with a change of ≥ 0.17 logCFU before and after BCG vaccination, although the rationale for this definition is not clear. *This definition is based on the standard deviation of multiple standard curves, in particular the inoculum dilution, used over multiple experiments with the same standardized batch of BCG. We have now added the information on the standard curves to the source data file and clarified the text in the manuscript (line 359-360) “defined as $\Delta \log \text{CFU (post (V3) - pre (V1))} < -0.17$, being the SD mean inoculum”.*

- Figure 2 is difficult to follow. Immune cell frequencies, plasma cytokine/chemokine markers, and cytokine/chemokines produced after in vitro BCG stimulation are presented in heat maps, each clustered by analyte and by participants that have been stratified into 6 groups. The authors observe ‘branching’ in the heat maps although the biological relevance is not clear. Reasons for reproducing sections of the heat maps in Figure 2B and 2C are not clear. *We thank the reviewer for raising these points and have revised the figure. We have now included the statistical significances in a new supplementary figure 4a-d, as adding the statistical significance to the heatmaps did not improve clarity. In addition, we have deleted the reproducing sections and adjusted the text accordingly.*

- The authors use an independent cohort of BCG re-vaccinated young adults (18-20yrs) in India to validate findings from their cohort of healthy Dutch adults. There are substantial differences between the Indian and Dutch cohorts, thus making comparisons and interpretation of the data between these two cohorts difficult. It is also not clear what value the immune profiling data from the Indian cohort add to the overall conclusions of the study, which are largely based on scRNA-seq data conducted only in the Dutch cohort.

In this manuscript we have used the Indian cohort solely as an independent validation cohort. We do not directly compare both cohorts but the functional and immune profiling data obtained in the Indian cohort validates our findings for the Dutch cohort, in particular the pre-existence of control prior to vaccination and the lack of control induction in a considerable part of the population, even in the re-vaccination setting.

- Immune profiling of samples from the Indian cohort are referenced in a submitted manuscript, and also presented in this manuscript, this creating the impression of duplication of data reported in the two manuscripts.

The Indian cohort consisted of 20 individuals with and 20 individuals without BCG-revaccination. In this manuscript we only describe these individuals in relation to the functional MGIA results with only major cell subset immune phenotyping profiles. The referenced submitted manuscript studies the effect of BCG-revaccination in individuals with or without revaccination in relation to innate vs. adaptive immune responses with heterologous bacterial, fungal and viral stimuli and a flow cytometry intracellular detection of cytokines for the different T-cell subsets.

- The data presentation in Figures 3-5 are not clear, with the text not legible as currently presented. *We thank the reviewer for pointing this out. We have the data available in high resolution but fear that the original format sent to the reviewer may not have been of optimal quality due to size and format restrictions of the platform. The present resubmission includes all figures in high resolution enabling maximal electronic magnification.*

- ScRNA-seq was conducted on PBMCs that were either incubated in media alone or stimulated for 4 hours with LPS. The rationale for analysis of gene expression in LPS-stimulated PBMCs is not clear; it is also unclear how gene expression in LPS-stimulated PBMCs provides further insight into mechanisms by which PBMCs restrict BCG growth.

We thank the reviewer for this critical, but valid, comment, also raised by reviewer 1, and would like to explain the choice for LPS in a bit more detail. First of all, we wanted to stimulate predominantly the myeloid compartment to assess monocyte reprogramming, and not reactivate antigen specific memory T- or B-cells. Secondly, we wanted to assess responses to a heterologous stimulation compared to the in vivo BCG stimulation. Thirdly, pilot experiments included both LPS and S.aureus stimulation for 4 hours and in these LPS induced a greater number of DEGs compared to the unstimulated comparator. LPS was thus selected as heterologous innate immune activator, to assess immune activation in these samples and compare to the functional capacity to control BCG outgrowth. Interestingly, most differences were already abundantly present in the unstimulated control samples, reflecting a different state of the cells, rather than an effect of LPS stimulation.

- The manuscript oscillates between focusing on trained immunity and mycobacterial growth inhibition, two themes that are not necessarily directly linked, thus detracting from clarity in the overall message of the paper.

We thank the reviewer for this comment. We would like to take liberty to refer to our previous work in which we have shown a relation between the functional mycobacterial growth assay and trained immunity. In that work we have shown that training did not only augment cytokine production but also increased the cellular ability to control mycobacterial growth. We have shown that CXCL10 produced by non-classical monocytes is a mechanistic marker of trained immunity and that the CXCL10-CXCR3 axis was critical in mycobacterial outgrowth control (Joosten, S.A. et al. Mycobacterial growth inhibition is associated with trained innate immunity. The Journal of clinical investigation 128, 1837-1851 (2018)). Thus we would contend that the two phenomena are indeed interlinked.

- The manuscript would benefit from revisions to reduce the text and make the manuscript concise with a more focused message. The figure legends are lengthy and difficult to follow. The figure legends include description and interpretation of the data, which should be in the Results section of the text rather than repeated in the figure legends.

As suggested by the reviewer we have gone through the legends critically and moved all references on interpretation of the data to the result section. We also shortened the introduction and improved the focus and quality of the manuscript with all comments received from the reviewers.

REVIEWER COMMENTS

Reviewer #1 (Remarks to the Author):

The revisions of this manuscript have been a great improvement. I thank the authors for their considered revision of the manuscript, the inclusions of the additional data/figures requested have enhanced the interpretability and contextualisation of the findings. There do remain some minor corrections and adjustments to be made but overall the article presents valuable new insights into potential mechanisms of BCG-mediated control of mycobacteria and the interplay between specific responses and heterologous/trained immunity responses.

Comments

Given a different strain (BCG Pasture) was used for the MGIA assays, please include discussion of the potential for the difference in strain between what was used for vaccination (Bulgaria/Russia) and MGIA assays (Pasture) to explain why measures of vaccine take did not discriminate between the MGIA defined functional groups

Please explain why different gating strategies were used for spectral (Indian cohort) and flow cytometry (Dutch cohort) where the same marker combinations were available (e.g. sup figure 3 shows exclusion of CD56+ cell from monocyte gates for the Dutch cohort (sup fig 3b) but not the Indian cohort (sup figure 3c). A comment should be added to the limitations on the potential impact different analysis methods had on the spectral/flow cytometry-based findings.

Minor Corrections:

Overall there are a number of typographical errors which will need correction (double spaces, species names with out space after the full top e.g. S.aureus).

In description of Figure 2h (lines 468-469), please amend statement "Increased classical CD14+CD16- monocytes prior to revaccination was present in individuals that were not able to control BCG growth even after revaccination." The data does not support this statement given the high level of variance across all groups.

The following sentence is unclear, please rephrase (lines 391-392) "Third, participants of the Indian BCG-revaccination could choose to be revaccinated or not, albeit the Dutch participants also volunteered to take part in the study."

There seem to be labelling issues on multiple figures (in both the PDF and word doc versions), please correct these:

- Fig 2 heatmaps are missing x-axis labels and volcano plot superscript text is overlaid on full-size text making it unreadable.
- Supplementary Figure 4 is missing scale bars (it appears they have not formatted correctly as the text for them is visible) and has errors with the formatting of labels which make them difficult to read.
- Please include scale bars for Figure 5F and supplementary Figure 8

Reviewer #2 (Remarks to the Author):

the authors have addressed the reviewers comments

Reviewer #3 (Remarks to the Author):

The authors have adequately addressed the comments raised by the reviewers. The authors have responded to my specific comments in a satisfactory manner, although please see additional comments below for consideration:

1. Check the data presented in Supplementary Figure 1a: IL-1b production after stimulation of PBMCs for 24 hr with S.aureus (lines 340-342 in manuscript). It is not clear why there is such a significant reduction in IL-1b production to S.aureus post-BCG vaccination in S.aureus non-responders.
2. Supplementary Figure 4: increase the font size of the text labels in each panel; the text is very difficult to read as currently labeled.
3. In general, the figures are still difficult to read, even after evaluating the source Word doc magnified at the maximum of 500%.

Reviewer #4 (Remarks to the Author):

In this study, authors aim to uncover immunological basis for responsiveness to BCG vaccination in two cohorts one from Netherlands and the second from India. The manuscript uses both a functional and a transcriptomic approach to address this question. Immunity induced by BCG is interesting as it is associated with reduced all-cause mortality in infants and in some studies increased responses to COVID-19 vaccines (although this remains a controversial topic). Mechanisms underlying these positive effects are believed to be related to “immune training”.

The overall conclusions of the study are that individuals who already have the ability to control M.Tb growth, those who acquire this after immunization, and those who never develop the ability have distinct immune profiles before and after LPS stimulation. This is not surprising given the different functional outcome.

Overall, the study address an interesting question but there are several concerns with the study design, data analysis and presentation, and conclusions outlined below:

1. The introduction is disconnected from the results/goals of the paper.
 - a. It is very focused on mechanisms by which BCG may exert its beneficial effects and trained immunity. However, none of the experiments in this paper deal with trained immunity, bio-energetics, or epigenetics
 - b. The introduction suggests that all those vaccinated gain ability to control bacterial growth 4-12 weeks post vaccination which is then lost at 1 year. However, this paper is focused on understanding who a large number of individuals do not respond to vaccination.
2. Study design/analysis
 - a. For the first comparisons, the authors randomly selected 42 participants based on response to Staph aureus after vaccination then reclassified them into prior controllers, acquired controllers and non-controllers with regards to BCG. Then they proceeded to draw major conclusions after sex differences and other comparisons. The appropriate approach would have been to randomly subset from the original 325 individuals into those categories.

Their data as it currently stands is potential biased by this flawed design.

- b. The authors describe a myriad of stimulation experiments as well as bacterial growth assays but those values are not well integrated into the analysis or the story. Their value to the study is unclear
- c. Line 174 – how long after addition of BFA or monnesin did the stimulation go on for?
- d. MIGA assay line 182 – which PBMC were used here? The BCG300 or the Indian cohort or both?
- e. For scRNA-Seq – it is unclear how many samples from each group were used. The text mentioned 5 samples from different donors and stimuli then the text mentions multiple batches with multiple samples. A clear experimental design needs to be submitted. This concern was raised during the prior review
- f. The methods used for DEG analysis from scRNA-Seq are not appropriate – they have been deemed to overestimate the number of DEG. It is best to use pseudobulk EdgeR-QLF was shown to be the best method in 2021

3. Results

- a. The sex differences in aim 1 are likely mediated by the rather unorthodox manner in which these samples were chosen – as stated in the summary, the authors should have run this analysis on all the samples, rather the ones preselected based on responses to an unrelated bacteria *Staph aureus*
- b. A list and markers used to delineate 73 subpopulations with a 10 marker panel should be provided- this is an unusually high number of subpopulations
- c. Statistics should be added into the figure – having to look up additional supplemental figures to understand if a difference is significant is tedious
- d. The significance of the Luminex data lines 366-378 is not clear
- e. The numbers in lines 391-393 do not match the methods section
- f. A UMAP with all the cells with clusters and conditions designated should be provided along with a bubble plot or violin plot of the marker genes so that the reader can assess how the populations were annotated
- g. Along with the comment about lack of clarity for the scRNA -seq design, can the authors confirm that they did not hash the individual samples and therefore cannot do statistics on the differences in subpopulation frequencies

REVIEWER COMMENTS

Reviewer #1 (Remarks to the Author):

The revisions of this manuscript have been a great improvement. I thank the authors for their considered revision of the manuscript, the inclusions of the additional data/figures requested have enhanced the interpretability and contextualisation of the findings. There do remain some minor corrections and adjustments to be made but overall the article presents valuable new insights into potential mechanisms of BCG-mediated control of mycobacteria and the interplay between specific responses and heterologous/trained immunity responses.

Comments

Given a different strain (BCG Pasture) was used for the MGIA assays, please include discussion of the potential for the difference in strain between what was used for vaccination (Bulgaria/Russia) and MGIA assays (Pasture) to explain why measures of vaccine take did not discriminate between the MGIA defined functional groups

We agree with the reviewer that subtle differences exist between strains of mycobacteria, including variant strains of BCG. The differences between both vaccination strains (Bulgaria and Russia) are minimal, however both strains still contain the RD3 region, which is absent in the Pasteur strain, used in the in vitro MGIA. RD3 is a small region, encoding approximately 10 antigens of the more than 4000 total antigens.

Vaccine take in vitro was measured using Mtb lysate (from H37Rv) for the 300BCG cohort or live BCG (TUBERVAC™, Serum Institute of India, Russian strain) for the Indian cohort, with T-cell cytokine production as read out, thus for the 300BCG cohort also not the identical strain used for vaccination, but containing RD3. Vaccine take was observed in all functional groups by assessing T-cell responses, which were detected in equal frequencies in all groups.

Thus, in the functional assay we have used a strain that lacked 10 antigens compared to the vaccination strain, however, RD3 does not contain any known immunodominant antigens. Moreover, as previously shown, the functional growth inhibition is an interplay between monocytes and T-cells, with the antigen specific component possibly in the T-cell compartment. Since the difference in antigenic repertoire is minimal (10 of 4000 antigens), we do assume that this is not the full explanation for the differences observed. Nevertheless, we have included this into the discussion of the manuscript (lines 695-701).

We frequently are questioned if the functional responses are strain specific, therefore we investigated growth inhibition capacities towards not only BCG but also pathogenic Mtb strains (H37Rv). We have used samples unrelated to this study, but consider growth inhibition over strains an important asset of the immune system. In, yet unpublished, studies on PBMCs from BCG vaccinated non-human primates, we have compared the MGIA responses to BCG (Pasteur) and Mtb (H37Rv) and observed a strong correlation (data plotted below, for reviewer only).

PBMC samples (n=84) from BCG vaccinated NHP's were used for a comparative analysis in the functional mycobacterial growth inhibition assay. Growth was measured for BCG and Mtb using PBMCs from the same animal, pre and post vaccination. Results are shown as logCFU for Mtb on the y-axis and BCG at the x-axis. A Spearman's rank correlation coefficient was calculated.

Please explain why different gating strategies were used for spectral (Indian cohort) and flow cytometry (Dutch cohort) where the same marker combinations were available (e.g. sup figure 3 shows exclusion of CD56+ cell from monocyte gates for the Dutch cohort (sup fig 3b) but not the Indian cohort (sup figure 3c). A comment should be added to the limitations on the potential impact different analysis methods had on the spectral/flow cytometry-based findings.

We thank the reviewer for the comment and agree that it should have been shown, as the gating strategy was equal for markers included in both panels. We now have adapted supplementary figure 3c to show consistency in the strategy followed. Furthermore we have added to the discussion (lines 708-712): "Fifthly, In this study we could not use the full potential of all accessible flow cytometry data as the different cohorts were measured on different analysers either by conventional flow cytometers or spectral analysers and with different marker panels. Despite overlapping markers, the choice of clones and fluorochromes was dependent on the analyser available, minimizing options for direct comparisons".

Minor Corrections:

Overall there are a number of typographical errors which will need correction (double spaces, species names with out space after the full top e.g. S.aureus).

We thank the reviewer for the accuracy and have corrected accordingly.

In description of Figure 2h (lines 468-469), please amend statement "Increased classical CD14+CD16- monocytes prior to revaccination was present in individuals that were not able to control BCG growth even after revaccination." The data does not support this statement given the high level of variance across all groups.

We agree with the reviewer that the variance is too large to state this. We have now amended the statement to: "Classical CD14+CD16- monocytes showed a large variance in all groups and there was no correlation between control state and frequency of this monocyte subset." (Lines 420-421)

The following sentence is unclear, please rephrase (lines 391-392) "Third, participants of the Indian BCG-revaccination could choose to be revaccinated or not, albeit the Dutch participants also volunteered to take part in the study."

We rephrased this sentence to: "Thirdly, participants of the Indian BCG-revaccination could choose to be revaccinated or not thereby introducing a selection bias. However the Dutch participants volunteered to take part in the BCG vaccination study, thus also recruiting individuals to receive a BCG vaccination, is potentially leading to a similar bias".

There seem to be labelling issues on multiple figures (in both the PDF and word doc versions), please correct these:

- Fig 2 heatmaps are missing x-axis labels and volcano plot superscript text is overlaid on full-size text making it unreadable
- Supplementary Figure 4 is missing scale bars (it appears they have not formatted correctly as the text for them is visible) and has errors with the formatting of labels which make them difficult to read.
- Please include scale bars for Figure 5F and supplementary Figure 8

We thank the reviewer for noticing these formatting issues and have checked all figures and corrected accordingly. Supplementary figures 4 a-d have been incorporated with figure 2.

Reviewer #2 (Remarks to the Author):

the authors have addressed the reviewers comments

We thank the reviewer.

Reviewer #3 (Remarks to the Author):

The authors have adequately addressed the comments raised by the reviewers. The authors have responded to my specific comments in a satisfactory manner, although please see additional comments below for consideration:

We thank the reviewer for the positive feedback and summarize below the actions taken to address the considerations.

1. Check the data presented in Supplementary Figure 1a: IL-1b production after stimulation of PBMCs for 24 hr with *S. aureus* (lines 340-342 in manuscript). It is not clear why there is such a significant reduction in IL-1b production to *S. aureus* post-BCG vaccination in *S. aureus* non-responders.

*Our participants were selected on the extremes of the spectrum of training responses to BCG vaccination. Trained innate immunity is expressed as the ratio, between pre and post vaccination samples, of cytokine production in response to an unrelated antigen, in this case *S. aureus*. All individuals were selected on a low vs high ratio of IL-1b production in response to *S. aureus* to*

identify non-responders and good responders. As result of selection on this ratio, differences in pre-vaccination cytokine production were identified for S. aureus but not for C. albicans or M.tuberculosis stimulation. The differences are thus the result of the selection process, albeit that was done on ratios over time and not absolute cytokine concentrations. However, and more importantly, if we do group on basis of the functional outcome, which is key to this study, we do not observe differences in S. aureus induced IL-1 β production, neither pre-nor post vaccination (Supplementary Figure 2b).

2. Supplementary Figure 4: increase the font size of the text labels in each panel; the text is very difficult to read as currently labeled.

In response to reviewer 4 we have adapted supplementary figure 4 and have implemented the statistical analysis to figure 2, thereby increasing the readability of the statistical results, but also paid attention to the readability of the fonts . The supplementary figure 4 has now changed to supplementary figure 5 as a schematic overview of the study has been inserted.

3. In general, the figures are still difficult to read, even after evaluating the source Word doc magnified at the maximum of 500%.

we have tried to improve the readability to the best of our abilities and have increased font sizes where possible following the journal's guidelines.

Reviewer #4 (Remarks to the Author):

In this study, authors aim to uncover immunological basis for responsiveness to BCG vaccination in two cohorts one from Netherlands and the second from India. The manuscript uses both a functional and a transcriptomic approach to address this question. Immunity induced by BCG is interesting as it is associated with reduced all-cause mortality in infants and in some studies increased responses to COVID-19 vaccines (although this remains a controversial topic). Mechanisms underlying these positive effects are believed to be related to “immune training”. The overall conclusions of the study are that individuals who already have the ability to control M.Tb growth, those who acquire this after immunization, and those who never develop the ability have distinct immune profiles before and after LPS stimulation. This is not surprising given the different functional outcome.

Overall, the study address an interesting question but there are several concerns with the study design, data analysis and presentation, and conclusions outlined below:

We thank the reviewer for the careful evaluation of the manuscript and will reply to the concerns raised in the point-by-point reply below.

1. The introduction is disconnected from the results/goals of the paper.

a. It is very focused on mechanisms by which BCG may exert its beneficial effects and trained immunity. However, none of the experiments in this paper deal with trained immunity, bio-energetics, or epigenetics

In this manuscript we analyse if two BCG (re)vaccination cohorts are capable to control mycobacterial growth in the functional mycobacterial growth inhibition assay. The individuals from the larger Dutch cohort were selected on base of their IL-1 β responsiveness to S.aureus as a hallmark of trained immunity. Previously, we have observed that increased growth control was related to markers of trained innate immunity, and to confirm these findings we now selected individuals with strong vs poor trained innate immunity induction upon BCG vaccination. We identified, based on our functional assay, three different BCG controlling groups. To our surprise the BCG control capacity of the individuals did not correlate with the induction of trained immunity, as measured by S.aureus induced IL-1 β production.

It would be interesting to investigate in future studies if there is a better correlate of trained immunity that ties the functional growth inhibition assay together. Furthermore methylation profiles were measured for these samples but due to relatively small sample size in the different BCG controlling groups the results were inconclusive. The identification of metabolic programming as discriminator between functional groups prompts to expand the different BCG controlling groups so methylation profiles with enough statistical power can be examined.

Interestingly, in our study the scRNAseq pathway analysis revealed dominant metabolic pathways, like glycolysis and oxidative phosphorylation, as key pathways differentiating the functional groups. This lead will be followed with flow cytometry based methods to study metabolic changes on a single cell level, but that is beyond the scope of this paper.

Based on the comments by the reviewer we have adapted our introduction to better emphasize the goal of the study and to increase the coherency between introduction and results.

b. The introduction suggests that all those vaccinated gain ability to control bacterial growth 4-12 weeks post vaccination which is then lost at 1 year. However, this paper is focused on understanding who a large number of individuals do not respond to vaccination.

We thank the reviewer for pointing out that we were not clear in this part of the introduction. We refer to our previous work where we tested a small cohort of BCG vaccinees pre and several timepoints post vaccination in the functional MGIA. The data showed that not all participants had a peak response for BCG control at the same timepoint. We had included 16 individuals of which two had a peak response at week 4 post vaccination, 8 at week 8 and 6 at week 12. Interestingly, and that was where the text was referring to, most of them had lost the capacity to control already partially at the next time point, following the peak response, which was only 4 weeks later. Retrospectively, if we would not have tested these samples longitudinally but only tested samples at week 12 as done in this study, we would also have identified individuals that did not control BCG growth at week 12 post vaccination because the peak response occurred earlier after BCG vaccination.

We have now revised the last paragraph of the introduction taking these points into account.

2. Study design/analysis

a. For the first comparisons, the authors randomly selected 42 participants based on response to Staph aureus after vaccination then reclassified them into prior controllers, acquired controllers and non-controllers with regards to BCG. Then they proceeded to draw major

conclusions after sex differences and other comparisons. The appropriate approach would have been to randomly subset from the original 325 individuals into those categories. Their data as it currently stands is potential biased by this flawed design.

*We thank the reviewer for careful looking into the study design. The selection of the 42 participants was balanced for distribution of sex and indeed based on the *S.aureus* response as measure of trained immunity (see source data file, tab Fig S2&S3). Previous work showed the relation between MGIA and trained immunity in cohorts of BCG vaccinees, individuals recently exposed to *Mtb*, TB infected individuals and TB patients (JCI **128(5)**, 1837-1851 (2018)). In the current study we aimed to further investigate the effect of trained immunity on immune cells and the mycobacterial growth inhibition assay (MGIA) and therefore selected these 42 participants with a known trained immunity profile. As trained immunity is evident in approximately 50% of healthy volunteers after BCG vaccination we selected both responders and non-responders to *S. aureus* as hallmark of training, to allow comparison of different trained immunity stages on functional MGIA control.*

For all individuals and timepoints tested for MGIA also scRNAseq was performed, in unstimulated and LPS stimulated samples. As MGIA and scRNAseq samples originate from the same vial of PBMCs this enabled us to integrate the analysis. The MGIA results determined the classification into the three groups of BCG control, with the distinct pathway profiles based on the scRNAseq data.

We appreciate the reviewer's comment and have adapted the introduction to explain the link between trained immunity, the MGIA results and the scRNAseq in more detail. Furthermore we have added supplementary figure 1 with the schematic overview of the design and workflow of this study.

b. The authors describe a myriad of stimulation experiments as well as bacterial growth assays but those values are not well integrated into the analysis or the story. Their value to the study is unclear

Mycobacterial growth inhibition data are the basis for the analysis presented here, those were performed on all samples and the outcomes of functional control have been used to evaluate all other data. Primarily, this was the scRNAseq performed on all samples, but secondary analyses were performed against stimulation data, as surrogates of trained immunity and vaccine take.

The stimulation experiments in supplementary figure 2 show the original data that underlies the selection of the 42 Dutch participants as was previously suggested by the reviewers. By plotting the same stimulation data in different ways we show that the trained immunity status does not correlate with the BCG control grouping and therefore we believe there is no bias in our selection of the 42 participants.

*Stimulation data in supplementary figure 3 is shown as measure of vaccine take in all participants. For the Dutch cohort BCG scarring is used as measure for vaccine take as is the IFN γ response to *Mtb* lysate in a seven days PBMC coculture. BCG scarring is equal between all BCG controlling groups (already – acquired and no control) and IFN γ responses against *M.tuberculosis* stimulation pre and post vaccination were detected, making differences in vaccine take between the functional control groups unlikely.*

In the Indian cohort, significant differences in the CD4+ IFN γ response to BCG pre vs post revaccination are observed, but are equal between the BCG controlling groups indicating no bias towards vaccine take.

c. Line 174 – how long after addition of BFA or monnesin did the stimulation go on for?

In lines 177-180 we state the total time of stimulation is 12 hours and 2 hours after start of coculture, BFA and monensin were added resulting in a stimulation time of 10 hours in the presence of BFA and monensin.

d. MIGA assay line 182 – which PBMC were used here? The BCG300 or the Indian cohort or both?

This comment refers to the method section describing the MGIA which was run, in an identical manner, on frozen PBMC's from both cohorts. Results for the Dutch (300BCG) cohort are shown in figure 1a and b and for the Indian cohort in figure 2d.

e. For scRNA-Seq – it is unclear how many samples from each group were used. The text mentioned 5 samples from different donors and stimuli then the text mentions multiple batches with multiple samples. A clear experimental design needs to be submitted. This concern was raised during the prior review

We thank the reviewer for the comments and understand the confusion with the phrasing of the different steps. We have therefore added a schematic overview to summarize the process of sampling, pooling and analysis. This is added to the supplementary information as supplementary figure 1. Furthermore we have revised the introduction and our material and methods section (lines 116- 124; 258-268).

Briefly, all 84 samples from the Dutch cohort were randomly tested in the MGIA in three separate runs. Samples pre and post BCG vaccination belonging to the same individual were always paired within the run. All samples used for MGIA were processed for scRNASeq, leading to a total of 168 samples either stimulated with LPS as heterologous stimulus for 4 hours or left unstimulated.

Firstly scRNAseq samples originate from the same PBMC vial as the cells for the MGIA. We have run the MGIA in 3 independent experiments and therefore 3 batches of scRNAseq samples were processed and sequenced. Each batch of RNA samples consisted of 10 to 12 pools of 3 to 5 samples. Within the pool an individual was unique and samples (varying in timepoint and stimulation) were randomly distributed over the pools. After sequencing each pool was demultiplexed based on genotype data. The cell distribution was visualized to check if there're any batch/donor effects before any downstream analysis. From this point onwards the standard workflow for scRNAseq analysis was used (Seurat v4, Cell 2019).

f. The methods used for DEG analysis from scRNA-Seq are not appropriate – they have been deemed to overestimate the number of DEG. It is best to use pseudobulk EdgeR-QLF was shown to be the best method in 2021.

We greatly appreciate the reviewer for raising this crucial question regarding the analysis of differentially expressed genes (DEGs) in single-cell RNAseq data. In our study, we used the standard workflow, specifically Seurat version 4, which was published in Cell in 2019. It is

important to acknowledge that, to date, there is no single absolute and accurate method for single-cell RNAseq analysis. As a result, this remains an open question for researches to address and explore further.

However, in response to the reviewer's valuable suggestion, we conducted a comparison between pseudobulk (edgeR-QLF) and single-cell (Seurat) analysis. For example, when comparing the "already controlling group" and "no control group" in monocytes, before BCG vaccination with LPS stimulation (Figure3), we found 159 up-regulated and 180 down-regulated genes using single-cell (Seurat) workflow, while 345 up-regulated and 478 down-regulated genes were identified using pseudo-bulk (edgeR-QLF) method. This result suggests that the single-cell (Seurat) workflow did not overestimate the number of DEGs compared to pseudo-bulk (edgeR-QLF) method in our data.

Furthermore, both methods yielded comparable results, with 141 shared genes between them showing consistent directions of effect size (logFC) as depicted in the figure for the reviewer below.

Effect size (logFC) comparison between pseudobulk (edgeR-QLF) and single-cell (Seurat) method. X axis is the effect size from edgeR-QLF method and y axis is the effect size from Seurat. Red line is the diagonal line.

Next, we conducted further tests to assess whether the enriched pathways identified by the single-cell (Seurat) analysis (Figure 3d) could be replicated using the pseudobulk method. The table (below) displays the enriched pathways by pseudobulk method, demonstrating our ability to reproduce the identified enrichments. These results highlight that the key signals remain consistent across both methods, indicating that they are not influenced by the analytical methods. Therefore we decided to maintain the analysis as used throughout the manuscript but have included the analysis strategy selection in the study limitations (lines 712-715)

Table: Pathway enrichment analysis (GO: Biological Processes)

GeneSet	p	adjP	genes
GO_RESPONSE_TO_INTERLEUKIN_1	1,03E-07	9,49E-05	PSME2:PSMB10:CYBA:UBB:UBA52:CD40
GO_INTERLEUKIN_1_MEDIATED_SIGNALING_PATHWAY	5,53E-06	1,69E-03	PSME2:PSMB10:UBB:UBA52
GO_RESPONSE_TO_TUMOR_NECROSIS_FACTOR	2,89E-05	5,06E-3	PSME2:TRADD:PSMB10:CYBA:CD40
GO_TUMOR_NECROSIS_FACTOR_MEDIATED_SIGNALING_PATHWAY	4,14E-05	6,62E-3	PSME2:TRADD:PSMB10:CD40
GO_RESPONSE_TO_INTERFERON_GAMMA	7,77E-05	9,84E-03	IFITM2:MT2A:CD40:HLA-C
GO_INTERSPECIES_INTERACTION_BETWEEN_ORGANISMS	8,81E-05	1,05E-02	S100A9:IFITM2:ISG20:PSMB10:UBB:UBA52:LGALS1
GO_RESPONSE_TO_TYPE_I_INTERFERON	1,95E-04	1,72E-02	IFITM2:ISG20:HLA-C

3. Results

a. The sex differences in aim 1 are likely mediated by the rather unorthodox manner in which these samples were chosen – as stated in the summary, the authors should have run this analysis on all the samples, rather the ones preselected based on responses to an unrelated bacteria *Staph aureus*

We thank the reviewer for the comment but it is not feasible to run MGIA's and scRNAseq on all 365 participants. We selected participants based on their trained immunity profile upon BCG vaccination, but ensured an equal sex distribution in all groups. We show in our source data (tab FigS2&3) and in the figure below for the reviewer that the sex distribution for the full cohort as well as across the good and the poor responders was balanced. We therefore believe the sex differences in the functional read-out are not biased by the selection of the participants.

Distribution of sex for the total cohort and divided for the trained immunity grouping based on the S.aureus IL-16 response. The number of individuals are plotted on the Y-axis with males in black and females in grey. Good responders: 11 males, 10 females; poor responders: 12 males, 9 females.

b. A list and markers used to delineate 73 subpopulations with a 10 marker panel should be provided- this is an unusually high number of subpopulations

*In the initial Dutch 300BCG cohort flow cytometry data was acquired using 6 different 10-color panels covering a broad spectrum of 73 subpopulations of immune cells, which was published in Cell reports **17**, 2474-2487 (2016) with a schematic overview of subsets in supplementary figure 5 and Cell Reports **33**, 108387 (2020). However, for this manuscript we have only used data from five of the available panels and have therefore adapted the materials and methods section to be accurate on the data shown. We also added the information on the panels used in the supplementary table 2.*

c. Statistics should be added into the figure – having to look up additional supplemental figures to understand if a difference is significant is tedious.

We appreciate the reviewers comment and have rebuilt the figure with the statistics directly parallel to the heatmap. We have extensively tried to add the statistics to the heatmap but the many comparison that can be made, i.e. pre/post vaccination within the group and between groups, impeded a clear and readable visualization when combining the information on expression or concentration and the statistics.

The revised figure 2a,b,c is now included in the manuscript.

d. The significance of the Luminex data lines 366-378 is not clear

*In our previous study (JCI **128(5)**, 1837-1851 (2018)) we had associated increased growth control with the presence of specific cytokines and chemokines in the supernatants of the MGIA assay, including CXCL9, 10 and 11. To investigate the presence of these and other chemokines in the supernatants of the current study we performed Luminex using the same reagents.*

We have now included this introduction prior to describing the Luminex data into the results section of the manuscript.

e. The numbers in lines 391-393 do not match the methods section

We thank the reviewer for spotting this inconsistency and have corrected the timepoint post vaccination for the Indian cohort to 10-12 weeks throughout the manuscript .

f. A UMAP with all the cells with clusters and conditions designated should be provided along with a bubble plot or violin plot of the marker genes so that the reader can assess how the populations were annotated

In this manuscript we only analysed the basic immune cell subsets and have shown the gating strategies in the supplementary figure 4. The flow cytometry analysis of the Dutch cohort was performed using five different 10-color panels and was therefore not suitable for high dimensional analysis like UMAP or bubble plots. The Indian cohort was measured using a large spectral flow

panel but given the relatively small group sizes refrained from extensive analysis as this was beyond the scope of the manuscript. Besides the supplementary table 1 with all information on the spectral flow panel we have now also included a supplementary table 2 with all information on the four panels and clones used for the Dutch cohort flow cytometric analysis.

g. Along with the comment about lack of clarity for the scRNA -seq design, can the authors confirm that they did not hash the individual samples and therefore cannot do statistics on the differences in subpopulation frequencies.

We sincerely thank the reviewer for his/her comments and we would like to clarify that we did not hash-tag individual samples in our study. Instead, we employed a random selection approach, where we chose the samples from four different conditions and allocated them into different pools. Within each pool, cells from 3 to 5 individuals were combined for single-cell RNA sequencing. Importantly, we didn't merge these individuals into a single sample during downstream analysis.

*To demultiplex individual cells and avoid any ambiguity, we used the widely used method called "souporcell" [Science, **376**: 6594 (2022), Nat Commun **13(1)**: 6149 (2022), Nat Commun **13(1)**: 3267 (2022)]. Souporcell utilizes genotype data from SNP calling to assign each cell to its corresponding donor and identify any doublets. This allowed us to retain the donor information for each cell, enabling us to conduct statistical tests as presented in the manuscript.*

REVIEWERS' COMMENTS

Reviewer #1 (Remarks to the Author):

The authors have addressed reviewers comments

Reviewer #3 (Remarks to the Author):

The authors have given thoughtful consideration to each of the reviewers' comments and made appropriate revisions to their revised manuscript. I do not have further comments.

This reviewer was also asked to assess the authors response to the concerns of reviewer 4 who is not available for comment this round.

I reviewed the comments from Reviewer 4 and the author's responses. I think the authors have responded appropriately and adequately to Reviewer 4's comments. I think there may be differences in opinion between the authors and Reviewer 4 regarding how the overall study should have been designed (i.e., the approach to selecting participants for further study), but it's not feasible for the authors to repeat the entire study and I don't think the approach that the authors used invalidates any of the findings they report in this paper. The authors have revised the manuscript to be more clear on the rationale and approach to the study, as well as study limitations, thus I feel that they have done everything they can to sufficiently address the points raised by Reviewer 4.